# Technological impact, generality, and complementarity of artificial intelligence patents: Evidence from Samsung Electronics

**Sangrok Lee, Taehyun Jung**[ORCID]*

Graduate School of Technology & Innovation Management, Hanyang University, Seoul, Republic of Korea

* tjung@hanyang.ac.kr

## Abstract

This study examines the extent to which artificial intelligence (AI) technologies are more pervasive and technologically complementary than non-AI technologies. Drawing on the literature on general purpose technologies (GPTs), we hypothesize that AI technologies exhibit greater pervasiveness—reflected in higher technological impact and broader generality—and stronger technological complementarity than non-AI patents. Using a firm-level analysis of patents filed by Samsung Electronics between 1982 and 2018, we find that AI patents demonstrate significantly higher impact, wider generality, and stronger systematic complementarity compared to matched non-AI patents. These results are robust across alternative matching specifications and multiple operationalizations of the key variables. Our findings offer firm-level empirical supporting AI's GPT-like characteristics and shed light on how AI-driven innovations may generate sustained incentives for technological advancement, even amid processes of creative destruction. The paper concludes by discussing key theoretical, managerial, and policy implications.

## 1. Introduction

The scale and speed of innovation in artificial intelligence (AI) have been remarkable in recent years, prompting growing recognition of AI's profound effects on economic growth, industry, scientific research, and the creative arts [1–9]. What distinguishes AI from other classes of technologies is its exceptional ability to generate follow-on innovations, its broad applicability across technological domains, and its capacity to create synergies when combined with other technologies—all of which are often discussed as characteristics associated with general-purpose technologies (GPTs) [10]. This study empirically examines whether these GPT-like features can be observed in AI patents within a single firm context.

Examining the GPT nature of AI is crucial for assessing its macroeconomic impact and for understanding innovation dynamics at the firm and industry levels, as suggested by Schumpeterian growth theory [11–13]. For instance, if AI can stimulate a

**Data availability statement:** All relevant data are within the paper and its Supporting information files.

**Funding:** The author(s) received no specific funding for this work.

**Competing interests:** The authors have declared that no competing interests exist.

higher volume of follow-on innovations in non-AI application sectors, its contribution to economic growth becomes multiplicative, and the competitive dynamics within AI research are likely to shift—favoring sustained innovation races rather than one-off technological leads. Notably, because GPT-type innovations were not explicitly modeled in their framework, Aghion and Howitt [13] themselves call for further research into the dynamics and implications of general-purpose technologies.

This paper responds to that call by offering firm-level empirical evidence. Building on prior studies [10,14–16], we examine three characteristics commonly associated with GPTs:

1) do AI technologies generate substantial downstream innovations (i.e., high technological impact)?

2) are they applicable across a wide range of sectors (i.e., broad generality); and

3) do they create systematic synergies with existing technologies (i.e., strong complementarity)?

While some existing studies have explored whether AI exhibits GPT-like features [4,17–19], most focus on macro-level patterns or rely on survey-based productivity measures. Few explicitly compare AI innovations to non-AI counterparts using firm-level technological data, leaving open critical questions about the intrinsic characteristics of AI as an innovation driver.

To address these questions, this study analyzes patent data filed by Samsung Electronics between 1982 and 2018. While prior research has employed firm-level surveys [5,20] or patent data at an aggregate level [21,22], we utilize firm-level patent data, which offers two key advantages: (1) the flexibility to construct fine-grained, innovation-specific indicators, and (2) comprehensive coverage across a wide range of technological domains. These strengths make patent data particularly well-suited for analyzing the characteristics and evolution of AI technologies—capabilities that survey-based approaches generally lack. To identify AI-related patents, we use the USPTO Artificial Intelligence Patent Dataset (AIPD), which employs machine learning techniques to classify patents into eight distinct AI technology categories [23].

Rather than including a wide sample of firms, we focus on a single-firm case study of Samsung Electronics. This design enables a more precise and internally consistent analysis, helping to minimize unobserved heterogeneity—such as variation in firm strategy, organizational structure, or geographic dispersion of patent citations—that would complicate a multi-firm research design. We acknowledge that a single-firm focus may limit generalizability to some extent. However, Samsung Electronics represents an analytically valuable case: it is a global market leader across multiple sectors—including electronics, semiconductors, and telecommunications—and maintains a robust R&D portfolio supported by numerous international collaborations. These features suggest that the patterns observed in Samsung's AI-related patenting activity may not be entirely idiosyncratic. Moreover, while Samsung is highly active in AI development, it is not a dedicated AI firm. This characteristic offers an analytical advantage: it allows us to draw broader insights into how AI is integrated and

deployed across a diversified innovation portfolio, rather than focusing on a narrow, AI-specialist context. A more detailed justification of our case selection is provided in the Empirical Strategy section below.

Our findings provide robust empirical evidence that AI patents exhibit significantly higher technological impact, broader generality, and stronger complementarity compared to carefully matched non-AI patents. These patterns are consistent across multiple econometric specifications and remain robust to alternative measurement approaches. Heterogeneity analyses further reveal that these effects are particularly pronounced in specific AI subfields—most notably computer vision, speech recognition, and machine learning.

These results contribute to ongoing debates about AI's technological nature and its broader economic implications. By offering patent-level, firm-based evidence of AI's distinctive innovation dynamics within a manufacturing-oriented global leader, our study complements existing macro-level research and sheds light on the mechanisms through which AI achieves widespread applicability and sustained technological influence. The findings suggest that AI innovations display characteristics consistent with transformative, broadly applicable general-purpose technologies, although we emphasize that realizing their full economic potential also depends on complementary organizational and institutional developments [14,17].

The remainder of the paper is structured as follows. The next section reviews the literature and develops an integrated framework connecting AI's technological characteristics to observable patent-level outcomes. The Methodology section introduces our empirical context and describes the data, detailing the coarsened exact matching methodology and regression specifications. The Results section presents the empirical findings on AI's impact, generality and complementarity within Samsung's portfolio. This paper concludes with a discussion of theoretical and practical implications, along with directions for future research.

## 2. Theoretical backdrop & hypothesis development

Given AI's expanding applicability and innovation-enabling potential, a growing body of scholarship has argued that AI constitutes a contemporary GPT, comparable to historical examples such as electricity, the steam engine, and computing technologies [4,5,17,20]. More recent empirical studies have begun to systematically assess AI's GPT characteristics using patent-based indicators. For example, Hötte et al. [15] compare multiple patent-based definitions of AI and find that AI consistently exhibits high rates of growth, technological generality, and complementarity across specifications—key hallmarks of GPTs. Similarly, Dong et al. [24] show that firms specializing in GPT-related R&D experience greater cross-industry expansion in supply chains, with these effects moderated by the breadth and depth of firms' technology portfolios.

Taken together, these theoretical and empirical insights provide a strong foundation for viewing AI as a transformative and broadly applicable technology, or GPT. The GPT perspective provides a useful lens for conceptualizing AI as a foundational technology that catalyzes innovation across diverse technological domains [25]. Agrawal et al. [26] emphasize AI's capacity to reduce the cost of prediction, thereby enabling new decision-making processes, organizational practices, and business models. Similarly, Furman and Seamans [27] highlight AI's productivity-enhancing effects in sectors such as transportation, healthcare, and education. Together, these studies suggest that AI can contribute to long-run economic growth by increasing total factor productivity and stimulating innovation across a broad set of industries.

To examine how AI innovations may exhibit distinctive technological characteristics, we draw on the defining features of GPTs: continuous technological improvement, pervasiveness across a wide range of uses, and innovational complementarities with existing and future technologies [10,28]. Given the well-documented rapid technological advancement of AI, this study focuses on the latter two dimensions—pervasiveness and complementarity. In the following sections, we develop hypotheses related to each dimension.

### 2.1. Is AI technology pervasive? AI's technological impact and generality

If AI technologies possess distinctive innovation-enabling characteristics, they should yield greater downstream technological development compared to non-AI technologies. For instance, AI algorithms developed for smartphone camera

enhancement have enabled downstream innovations in medical imaging, autonomous vehicle perception, and industrial quality control.

We identify three key mechanisms through which AI innovations may demonstrate higher impact and generality. First, AI technologies are more likely to result in productive and valuable outputs. Firm-level evidence shows that AI adoption significantly boosts productivity and innovation. For example, Czarnitzki et al. [5] find that AI adoption leads to substantial productivity gains across multiple industries, suggesting that AI technologies give rise to high-value downstream applications. Similarly, Rammer et al. [20] report that AI-adopting firms demonstrate increased innovation output. Although international diffusion studies (e.g., Jiang et al. [29]) show disproportionate follow-on development for AI technologies, these findings may conflate technological influence with strategic or organizational factors.

Second, AI facilitates novel knowledge recombination, enabling connections across disparate domains. Technological innovation often emerges through recombining existing knowledge elements [30], and AI is particularly suited to identify novel patterns across diverse fields. Rammer et al. [20] emphasize that AI's capacity to span disciplinary boundaries allows for cross-domain knowledge integration. In drug discovery, for example, AI systems recombine insights from chemistry, biology, and pharmacology to explore previously inaccessible knowledge in chemistry alone [31].

Recent empirical studies offer further nuance. Qin et al. [32] document an inverted U-shaped relationship between knowledge coupling and generality in AI patents: moderate coupling facilitates broad reuse by balancing novelty and comprehensibility, while excessive coupling introduces complexity that may limit applicability. This suggests that optimally structured AI innovations can serve as foundational platforms for subsequent technological development.

Third, AI acts as a convergence enabler, bridging previously disconnected technological domains. Using network and clustering analyses, Lee et al. [33] show that AI functions as a hub technology, increasingly co-classified with diverse technological areas. Since 2012, its convergence intensity has accelerated, expanding beyond electronics into healthcare, finance, logistics, and advanced manufacturing. This bridging role implies that AI innovations can spawn applications in varied domains, each leading to further downstream developments.

Drawing on these arguments, we propose the following hypotheses.

**Hypothesis 1:** *All else equal, AI patents will exhibit a higher technological impact than comparable non-AI patents.*

**Hypothesis 2:** *All else equal, AI patents will exhibit higher technological generality than comparable non-AI patents.*

## 2.2. Is AI technology complementary?

Innovation complementarity, in the context of GPTs, refers to the emergence of novel innovations when GPTs are combined with other, non-GPT technologies. Two technologies are considered complementary when their joint application creates greater value than their independent use [34].

We argue that AI technologies exhibit high levels of complementarity because they often function as "method technologies"—enhancing or enabling other technologies rather than operating in isolation. For example, machine learning algorithms systematically improve image sensor performance; natural language processing augments user interface technologies; and computer vision strengthens robotic control systems. Unlike generic general-purpose components that may combine randomly with various technologies, AI's functional nature implies non-random, systematic pairing with specific complementary technologies.

This complementarity is further reinforced by firms' strategic behavior. Firm-level studies suggest that the value of AI arises primarily through systematic integration with existing technological assets. Hoffreumon et al. [35] find that firms often develop internal AI capabilities while adopting external AI solutions—demonstrating complementarity rather than substitution. In this view, effective AI deployment is not a matter of broad application but of deliberate coupling with core

competencies. For Samsung, AI's integration with strengths in semiconductors, telecommunications, and consumer electronics exhibits patterns consistent with strategic alignment rather than random recombination.

Additionally, international diffusion studies suggest that AI functions as a hub technology within global knowledge networks [29], exhibiting consistent and systematic co-occurrence with specific technological domains. This pattern contrasts with technologies that appear in diverse contexts without generating strong or sustained synergies. AI's consistent pairing behavior supports its role as a complementary enabler rather than a standalone platform.

These insights motivate our third hypothesis:

*Hypothesis 3:* All else equal, AI patents will be higher in technological complementarity than comparable non-AI patents.

## 3. Methodology

### 3.1. Empirical strategy

We employ patent data as our primary data source because they offer a rich, expert-validated, and systematically structured record of technological innovation. Patents enable a detailed examination of the characteristics, impacts, and interrelationships of AI technologies over time. Moreover, they have been widely used in prior studies examining similar research questions [21–23,36]. At the same time, we remain attentive to the well-documented limitations of using patents as proxies for innovation, including strategic patenting behavior, variation in citation practices, and differences in patent quality [37–40]. These considerations inform our interpretation of the empirical findings and guide our methodological choices.

In this study, we adopt a single-firm study of Samsung Electronics for several compelling reasons. Samsung is a leading global technology firm with substantial technological and business diversity across a wide, yet coherent, range of sectors—including semiconductors, telecommunications, and consumer electronics. It combines this diversity with significant market power and consistently top-tier performance in R&D and patenting.

According to global statistics from 2018—the final year of our originating patent dataset—Samsung ranked 12th on the Fortune Global 500 list (second only to Apple among technology companies), second in patent filings at the USPTO (after IBM), and first in R&D investment [41,42]. In patenting, Samsung was followed by Canon, Intel, LG, TSMC, Microsoft, Qualcomm, Apple, Ford Motor, and Google, in that order [41]. In terms of R&D expenditure, Samsung led a group that included Google, Volkswagen, Microsoft, Huawei, Intel, and Apple [42].

Importantly, compared to peers such as Apple, Intel, or Canon, Samsung's broader product portfolio—spanning multiple global markets—offers a unique opportunity to observe how AI technologies diffuse across diverse technological domains. At the same time, Samsung's research operations are largely centralized in Suwon, South Korea, even as the firm competes internationally. This organizational structure helps reduce internal heterogeneity, enhancing the reliability of firm-level analysis.

In summary, Samsung represents both a prototypical and a frontier technology firm, offering a robust setting for examining the technological impacts and generality of AI. We argue that Samsung satisfies case selection criteria suggested by qualitative methodologists [43,44], while also mitigating the confounding factors that may arise in multi-firm or multi-country studies. Moreover, far from limiting generalizability, Samsung's multi-product, multi-market structure strengthens the relevance of our findings across different technological and industrial contexts.

This study builds upon an established body of research that has used Samsung as a focal firm [45–48], extending the analysis into the emerging and strategically critical domain of AI-driven innovation.

One may argue in favor of more generalizable, large-sample designs—such as firm-level panel datasets with fixed-effect models or simple dummy-variable controls for firm heterogeneity—to study AI's technological impact. However, we find such approaches unsuitable for our research objectives for several reasons. First, the use and citation of patents are often geographically or organizationally bounded [49–51], which can introduce bias when estimating the generality and

technological value of AI innovations across diverse contexts. While matched control group methods [50,51] can partially mitigate such biases, they present substantial challenges in ensuring robustness, precision, and comparability [52,53]— particularly given the high-dimensional, rapidly evolving nature of AI technologies. Furthermore, implementing such controls can introduce considerable complexity and noise into the research design, making it difficult to capture the nuanced, longitudinal evolution of AI technologies within a real-world firm setting.

Second, the utilization of prior innovations is often influenced by non-technological factors, such as institutional affiliations or social relationships between originators and users [54,55]. Strategic considerations or positional advantages may also shape citation and adoption behavior [40,56]. However, it remains unclear whether such biases systematically differ between AI and non-AI technologies, and we lack the ability to directly observe or control for the strategic motivations or relational dynamics among the actors involved. For example, differences in citing behavior may reflect organizational context rather than technological characteristics—AI patents from firms like Microsoft or Google may receive broader citations due to their central roles in software ecosystems, while Samsung's AI patents may be cited more narrowly despite comparable technological value.

Our single-firm design mitigates such confounding factors by comparing AI and non-AI patents within the same institutional, strategic, and organizational environment. This allows us to isolate the technological characteristics of AI innovations more reliably, without the added complexity of cross-firm heterogeneity.

Given these challenges, this study adopts a more tractable approach by focusing exclusively on originating patents filed by a single firm, thereby naturally controlling for many of these organizational and relational factors. While single-firm studies are sometimes criticized for limited generalizability, firm-specific idiosyncrasies, and reduced explanatory power due to unobserved heterogeneity, these concerns can be at least partially addressed by selecting a representative and analytically rich case. As argued above, we believe that Samsung Electronics serves this role effectively, offering a compelling setting for in-depth analysis of AI technologies within a global innovation leader.

### 3.2. Sample construction

As a first step in constructing our dataset, we collected all patents filed by Samsung Electronics from 1982 to 2018. We rely on the USPTO's bulk patent download service via the *PatentsView* database [57–59].

To ensure comprehensive and accurate identification of Samsung Electronics patents, we implemented a systematic assignee verification process combining multiple data sources. Initial extraction from USPTO *PatentsView* identified patents using various representations of the firm's name, including potential misspellings, alternative spellings, and overseas subsidiaries, to minimize false negatives. However, preliminary analysis revealed that USPTO's assignee standardization occasionally conflates distinct Samsung affiliates and unrelated entities with similar names. To address this issue, we cross-validated all Samsung Electronics patent assignments using the EPO PATSTAT database (2025 Spring edition), which provides independently standardized assignee information. After an extensive verification process (as described in the Appendix in S1 Dataset), we constructed a final dataset of 86,824 patents assigned to Samsung Electronics.

### 3.3. Matching sample

The core empirical strategy of this study is to compare the generality and technological impact of AI patents with those of non-AI patents. Regression models with control variables may yield biased estimates if the covariate distributions between AI and non-AI patents differ significantly. For instance, if AI patents are concentrated in a specific technology class where non-AI patents are sparsely represented, the regression model may extrapolate estimates beyond the observed data, introducing bias.

To address this concern, we employ coarsened exact matching (CEM) [60–62] to construct a control group of non-AI patents that are statistically comparable to the AI patents. CEM improves the comparability between treated and control

groups by balancing the distribution of covariates, thereby reducing model dependence. By matching units based on the exact values of coarsened covariates, CEM mitigates the risks associated with model misspecification [63] and has been shown to outperform alternative methods such as propensity score matching in terms of both bias reduction and estimation efficiency [64].

Consistent with prior studies [39,50,52,65,66], each AI patent, identified as described in Section 3.5, was matched with comparable non-AI patents according to the following criteria:

1) Technological similarity: patents must share at least one common Cooperative Patent Classification (CPC) code;

2) Filing year: patents must be filed in the same year or within a one-year window;

3) Inventor team size: patents must fall within the same category of number of inventors—less than 3, 4–6, 7–9, or more than 9 inventors; and

4) Patent strength: patents must belong to the same claim-count category—less than 6 claims, 6–10 claims, 11–15 claims, 16–20 claims, or more than 20 claims.

The matched sample consists of 10,695 patents—4,936 AI patents and 5,759 non-AI patents—representing approximately 12.3% of the original dataset of 86,824 Samsung Electronics patents. While the reduction in sample size is an inherent trade-off of the CEM approach, prioritizing balance and comparability between the treatment and control groups is essential for ensuring the validity of causal inference [61].

After matching, the number of AI patents has decreased from 6,915–4,936, reflecting the strictness of our matching criteria. Although this strictness results in the exclusion of some AI patents, it enhances the internal validity of the analysis by reducing the risk of confounding factors biasing our estimates of the effects of AI on technological generality and impact. Indeed, the post-matching sample exhibits improved balance, as shown in Table 1. We observed that statistically significant differences between AI and non-AI patents in both number of claims (mean = 18.577 for AI vs. 17.002 for non-AI, $p < 0.001$) and backward citations count (mean = 7.118 for AI vs. 6.514 for non-AI, $p < 0.001$) before matching had disappeared after matching, indicating that the matching procedure effectively balanced key covariates between the treatment and control groups.

### 3.4. Dependent variables

We measure the technological impact of each patent by the number of *forward citations* it receives within a five-year window from its filing date, following established practice in the literature [67–69]. We use a fixed citation window to ensure comparability across patents filed in different years. In our sample, the mean number of forward citations is 2.06, with a maximum of 559 citations for a single patent. (see Table 2).

We also compute several alternative measures to ensure robustness of our findings. First, we calculate forward citations excluding self-citations (citations from patents assigned to Samsung Electronics) to ensure that observed impact reflects broader technological influence rather than internal reuse. Second, we compute percentile ranks of forward citations within each filing-year × WIPO technology field combination to account for field-specific and temporal heterogeneity in citation propensities. Third, we calculate z-scores to provide standardized measures. Finally, we examine a 3-year citation window as an alternative to our baseline 5-year window. Results using these alternative measures are reported in the robustness checks section.

We use the generality measure proposed by Trajtenberg et al. [70] as our baseline dependent variable—hereafter referred to as *JTH generality*—as it has been widely adopted in the literature to capture the breadth of a patent's technological influence [71,72]. A higher generality score indicates that a focal patent is cited by subsequent patents from a more diverse set of technology classes, implying broader applicability.

**Table 1. Key statistics before & after matching (original CEM).**

| Covariates | Before matching | | After matching (1 to many) | |
|---|---|---|---|---|
| | **AI**<br>**Mean SD**<br>**T-stat** | **Non-AI**<br>**Mean SD**<br>**T-stat** | **AI**<br>**Mean SD**<br>**T-stat** | **Non-AI**<br>**Mean SD**<br>**T-stat** |
| Number of claims | 18.577<br>(8.383) | 17.002<br>(8.396) | 17.741<br>(6.551) | 17.603<br>(6.435) |
| | 14.969*** | | 1.101 | |
| Number of inventors | 3.179<br>(1.885) | 3.18<br>(2.092) | 3.08<br>(1.728) | 3.06<br>(1.817) |
| | −0.052 | | 0.569 | |
| Number of CPCs | 6.124<br>(4.908) | 6.706<br>(6.296) | 7.102<br>(5.001) | 8.807<br>(6.157) |
| | −7.492*** | | −15.554*** | |
| Backward citations | 7.118<br>(13.093) | 6.514<br>(9.498) | 7.069<br>(12.746) | 6.7<br>(12.16) |
| | 4.894*** | | 1.528 | |
| Non-patent references | 4.78<br>(13.48) | 2.143<br>(6.623) | 5.086<br>(14.361) | 3.736<br>(9.662) |
| | 28.405*** | | 5.77*** | |
| Observations | *6,915* | 79,909 | *4,936* | *5,759* |
| | 86,824 | | *10,695* | |

Notes: *** p<0.001, ** p<0.01, * p<0.05.

WIPO distribution

- before matching: distributed all 35 field,

- after matching: distributed 29 field except 6 field (id 15, 17, 18, 20, 24, 29).

We compute *JTH generality* using the Herfindahl-Hirschman Index (HHI) of concentration, based on the 4-digit CPC codes of citing patents. Only citations made within five years of the focal patent's filing are counted, to ensure a consistent citation window across time.

The generality of patent P is defined as:

$$JTH\ generality(P) = 1 - HHI(P) = 1 - \sum_{i=1}^{n} S(P_i)^2 \quad where\ S(P_i) = \frac{N(i)}{\sum_{i=1}^{n} N(i)}$$

Here, $S(P_i)$ represents the share of citing patents that fall under CPC code $i$, and $N(i)$ denotes the number of such patents citing the focal patent $P$. The summation runs over all CPC classes represented in the citing set. A lower HHI (and thus a higher generality score) reflects greater dispersion across technological fields.

We also use the Shannon entropy index as an alternative measure of diversity to triangulate our results. The Shannon entropy has been widely used in studies measuring scientific and technological diversity [73,74]. We define the *Shannon generality* of a focal patent P as:

$$Shannon\ generality(P) = -\sum_{i=1}^{n} S(P_i) \cdot log_2 S(P_i)$$

**Table 2. Sample statistics (N = 10,695).**

| | Mean | S.D. | Min | Max |
|---|---|---|---|---|
| Forward citations (5-yr) | 2.06 | 6.39 | 0 | 559 |
| Forward citations (5-yr, w/o self-citations) | 1.92 | 5.8 | 0 | 559 |
| Forward citations percentile (5-yr) | 50.48 | 26.48 | 2.59 | 100 |
| Forward citations Z-score (5-yr) | 0 | 1 | −1.74 | 26.36 |
| Forward citations (3-yr) | 1.02 | 3.08 | 0 | 229 |
| Forward citations percentile (3-yr) | 50.48 | 24.29 | 6.14 | 100 |
| Forward citations Z-score (3-yr) | 0 | 1 | −1.41 | 28.15 |
| JTH generality (5-yr) | 0.07 | 0.18 | 0 | 0.86 |
| JTH generality (5-yr, w/o self-citations) | 0.07 | 0.18 | 0 | 0.89 |
| Shannon generality (5-yr) | 0.17 | 0.44 | 0 | 3.53 |
| Shannon generality (5-yr, w/o self-citations) | 0.16 | 0.42 | 0 | 3.64 |
| Complementarity (5-yr) | 0.1 | 0.1 | 0 | 0.78 |
| Complementarity (10-yr) | 0.1 | 0.1 | 0 | 0.74 |
| AI patent | 0.46 | 0.50 | 0 | 1 |
| Number of claims | 17.14 | 8.41 | 1 | 160 |
| Number of inventors | 3.18 | 2.08 | 1 | 38 |
| Number of CPCs (ln) | 1.58 | 0.81 | 0 | 4.79 |
| Backward citations (ln) | 1.43 | 0.91 | 0 | 6.18 |
| Non-patent references (ln) | 0.63 | 0.87 | 0 | 5.59 |
| Days to grant | 1093.51 | 532.46 | 76 | 5490 |
| Patent family size (ln) | 1.23 | 0.63 | 0 | 5 |
| WIPO: Electrical machinery | 0.05 | 0.22 | 0 | 1 |
| WIPO: Audio-visual tech. | 0.15 | 0.35 | 0 | 1 |
| WIPO: Telecommunications | 0.06 | 0.24 | 0 | 1 |
| WIPO: Digital communication | 0.13 | 0.34 | 0 | 1 |
| WIPO: Computer technology | 0.2 | 0.4 | 0 | 1 |
| WIPO: Semiconductors | 0.15 | 0.36 | 0 | 1 |
| WIPO: Optics | 0.1 | 0.3 | 0 | 1 |
| WIPO: Other fields | 0.16 | 0.37 | 0 | 1 |

Unlike *JTH generality*, *Shannon generality* is not bounded between 0 and 1, allowing for greater differentiation—particularly among patents with highly dispersed technological influence. In our sample, the mean values of JTH and Shannon generality are 0.07 and 0.17, respectively. Shannon generality exhibits higher variance and a greater maximum value than *JTH generality* (see Table 2), suggesting that, on average, Samsung Electronics' patents receive forward citations from a moderately diverse set of technological fields. We also compute generality measures excluding self-citations to ensure our results are not driven by Samsung's internal citation patterns.

To capture *technological complementarity*, we adopt the complementarity measure proposed by Dibiaggio et al. [75], adapting it to the patent level. This measure assesses whether pairs of technologies co-occur more frequently than would be expected under random combination, given the observed distribution of technologies. Following prior literature (e.g., [75,76]), we assume that technology pairs that are more frequently combined within a patent are more closely related and technologically complementary than those that co-occur less often. An important advantage of this measure over simple co-occurrence counts, commonly used in prior studies (e.g., [15,77]), is that it explicitly controls for size effects across technology classes and allows for normalization, thus enabling more meaningful comparisons across technologies.

The standardized complementarity index between technology classes *j* and *k* denoted $\lambda_{jk}$, is calculated as:

$$\lambda_{jk} = \frac{C_{jk} - \mu_{jk}}{\sigma_{jk}}$$

where $C_{jk}$ represents the observed number of patents in a given year in which technology classes *j* and *k* co-occur. The terms $\mu_{jk}$ and $\sigma_{jk}$ denote the expected co-occurrence count and its standard deviation, respectively, under a random assignment derived from a hypergeometric distribution. We compute these expectations using rolling windows of five years and ten years to assess the robustness of the measure to different temporal horizons.

A positive value of $\lambda_{jk}$ indicates technological complementarity, meaning that the two technologies co-occur more frequently than expected by chance, whereas a negative value suggests substitutability, reflecting less frequent co-occurrence. Following Dibiaggio et al. [75], we normalize the complementarity index using a min–max standardization procedure. For each patent, we then compute the average complementarity score across all technology pairs contained within that patent.

Unlike Dibiaggio et al. [75], who aggregate complementarity at the firm level across entire patent portfolios, our approach constructs patent-level complementarity measures based on the specific combinations of technologies embodied in each patent. This allows us to compare technological complementarity across individual patents and across heterogeneous technological domains.

### 3.5. Independent variables and controls

Our primary independent variable is whether a patent contains elements of AI technology. Among various approaches to identifying AI-related patents, we adopt the USPTO's Artificial Intelligence Patent Dataset (AIPD) [23], which employs a machine learning–based classification method using the textual content of patent documents. The AIPD has been increasingly utilized in studies on AI innovation [15,21,22,25,78,79], reflecting its reliability and growing acceptance in the field. We define a binary variable, AI patent, coded as 1 if a patent is classified under any of the eight AIPD categories, and 0 otherwise. Summary statistics for this variable were presented in Table 2. In addition, we retain the full set of eight AI technology categories provided by AIPD (listed in Appendix 1 in S1 Dataset) and use them in robustness checks to examine potential heterogeneity in AI subfields.

We include a comprehensive set of control variables to account for patent characteristics that may influence a patent's generality or technological impact.

First, we control for the technological breadth of a patent by including the logarithm of *the number of CPC* codes assigned to each patent, as well as the logarithm of the number of claims, following prior studies [65,80,81]. A broader technology scope may increase the likelihood of a patent being cited across a wider range of subsequent inventions in both quantity and scope. For each patent, we counted unique CPC subgroup codes (full digit level) and applied a log transformation to correct for the right-skewed distribution. A higher number of CPC codes suggests that the invention spans multiple technological domains, which may reflect a more complex or broadly applicable innovation [30].

Second, we control for *inventor team size* by including the number of inventors listed on each patent. Larger inventor teams are often associated with more diverse expertise and knowledge recombination, which can affect the novelty and applicability of innovations [82]. In our sample, the average number of inventors per patent is 3.04 (Table 2).

Third, we include the *number of backward citations*—i.e., prior patents cited by the focal patent—as a control variable. Backward citations are a strong correlate of patent value and serve as a proxy for the depth and breadth of the technological foundations on which a patent builds [70]. To address the long-tail distribution of citation counts, we use the logarithm of the backward citation count.

Fourth, to account for the scientific knowledge base underlying a patent, we control for the number of *non-patent references* [83]. Patents that cite more scientific literature are often more basic in nature and may be applicable across a broader range of technologies [84]. We apply a log transformation to the NPL count to address its skewed distribution.

Fifth, we control for patent *family size*, defined as the number of patent offices where protection for the same invention was sought. Larger patent families indicate that the assignee considers the invention sufficiently valuable to warrant the costs of international patent protection [65,85]. We use the logarithm of family size to account for the right-skewed distribution. In our sample, the mean log-transformed family size is 1.23 (see Table 2).

Sixth, we include *days to grant*—the number of days elapsed between the patent application date and the grant date—to control for potential differences in examination processes and patent complexity. Longer examination periods may reflect greater technological complexity, novelty, or examination rigor [86]. The average examination period in our sample is 1,094 days, or approximately three years (see Table 2).

Seventh, we include application year dummies to control for potential temporal heterogeneity in patenting activity and innovation strategy. Changes in the macroeconomic environment, regulatory landscape, or internal R&D focus may affect patent characteristics over time. By using year-fixed effects based on the application year, we control for unobserved year-specific shocks that could influence both the likelihood of AI classification and patent outcomes. This approach ensures a more precise estimation of the effects of AI-related technologies, net of time-specific confounding factors.

Finally, to account for technological field-specific heterogeneity, we include dummy variables for WIPO technology fields as defined by Schmoch [87]. Of the 35 total fields, we focus on those prominently represented in Samsung Electronics' patent portfolio, including: electrical machinery, audiovisual technology, telecommunication, digital communication, computer technology, optics, and semiconductors. This approach allows us to control for systematic differences in generality and impact that may arise from the nature of the technological domain in which a patent is situated.

Correlations among independent and control variables are generally low, with most pairwise coefficients below 0.3 (Table 3). Variance Inflation Factors (VIFs) further confirm limited multicollinearity concerns, with all values below 3.74—well within acceptable thresholds.

## 3.4. Estimation models

To test Hypothesis 1, we estimate a regression model using the number of forward citations as the dependent variable:

$$Forward\ Citations_i = f(\beta_0 + \beta_1 AI_i + \gamma C_i + \varepsilon_i)$$

where *Forward Citations*$_i$ represents the count of forward citations for patent $i$, $AI_i$ is a binary indicator for AI patent, $C_i$ is a vector of control variables, $\varepsilon_i$ is the error term, and $f(\cdot)$ denotes the link function that accounts for the count nature of the dependent variable.

Forward citations are count data that exhibit substantial overdispersion (mean = 2.06, variance = 40.83, as reported in Table 2) and a high proportion of zero observations (53% of patents receive no citations within five years). Given these characteristics, we adopt a zero-inflated negative binomial (ZINB) regression model, alongside standard negative binomial (NB) models for comparison [88].

We begin by testing for overdispersion, and the diagnostic results clearly support the use of a negative binomial (NB) model over a Poisson model. Specifically, the dispersion parameter lnα is statistically significant (p < 0.001) across all models in Table 4, confirming substantial overdispersion.

To account for zero inflation in the ZINB models (Models 3–4), we include covariates in the inflation equation using the same set of control variables described previously, except for the WIPO technology field dummies and filing year dummies, which are excluded in the zero-inflation component. The inflation equation models the probability that a patent belongs to the "always-zero" group versus the count process group.

**Table 3. Correlation Matrix (N = 10,695).**

| | | 1 | 2 | 3 | 4 | 5 | 6 | 7 |
|---|---|---|---|---|---|---|---|---|
| 1 | AI patent (0 or 1) | 1 | | | | | | |
| 2 | Number of claims | 0.05 | 1 | | | | | |
| 3 | Number of inventors | 0 | 0.05 | 1 | | | | |
| 4 | Number of CPCs (ln) | −0.02 | 0.04 | 0.21 | 1 | | | |
| 5 | Backward citations (ln) | 0.01 | 0.02 | −0.04 | 0.02 | 1 | | |
| 6 | Non-patent references (ln) | 0.12 | 0.04 | 0.13 | 0.2 | 0.13 | 1 | |
| 7 | Days to grant | 0.09 | 0.09 | 0.01 | −0.06 | 0.01 | 0.21 | 1 |
| 8 | Patent family size (ln) | −0.01 | −0.05 | 0.03 | 0.2 | 0.1 | 0.38 | 0 |
| 9 | Electrical machinery | −0.06 | −0.01 | 0.02 | −0.02 | −0.01 | −0.02 | −0.04 |
| 10 | Audio-visual tech. | 0.01 | −0.01 | −0.08 | 0 | 0.01 | 0.04 | 0.07 |
| 11 | Telecommunications | 0.01 | 0 | −0.04 | −0.08 | −0.01 | −0.03 | 0.08 |
| 12 | Digital communication | 0 | 0.05 | 0.01 | 0.04 | −0.09 | 0.16 | 0.22 |
| 13 | Computer technology | 0.21 | 0.05 | −0.05 | −0.09 | 0.05 | −0.02 | −0.03 |
| 14 | Semiconductors | −0.12 | −0.04 | 0.1 | 0.2 | 0.05 | −0.12 | −0.19 |
| 15 | Optics | −0.08 | 0.01 | 0.01 | −0.07 | −0.06 | −0.08 | −0.06 |
| 16 | Other fields | −0.03 | −0.05 | 0.02 | −0.02 | 0.03 | 0.05 | −0.03 |
| | | 9 | 10 | 11 | 12 | 13 | 14 | 15 |
| 9 | Electrical machinery | 1 | | | | | | |
| 10 | Audio-visual tech. | −0.1 | 1 | | | | | |
| 11 | Telecommunications | −0.06 | −0.1 | 1 | | | | |
| 12 | Digital communication | −0.09 | −0.16 | −0.1 | 1 | | | |
| 13 | Computer technology | −0.12 | −0.21 | −0.12 | −0.19 | 1 | | |
| 14 | Semiconductors | −0.1 | −0.18 | −0.11 | −0.16 | −0.21 | 1 | |
| 15 | Optics | −0.08 | −0.14 | −0.08 | −0.13 | −0.16 | −0.14 | 1 |
| 16 | Other fields | −0.1 | −0.18 | −0.11 | −0.17 | −0.22 | −0.19 | −0.15 |

To test Hypothesis 2, we employ Tobit regression models to account for the censored nature of the dependent variables. The JTH generality measure is bounded between 0 and 1, while the Shannon generality measure is left-censored at zero. We estimate the following latent variable model:

$$\text{Generality}_i^* = \beta_0 + \beta_1 AI_i + \gamma C_i + \varepsilon_i$$

where generality$_i^*$ denotes the unobserved latent generality for patent $i$, $AI_i$ is a binary indicator for AI patent, $C_i$ is a vector of control variables, and $\varepsilon_i$ is the error term.

Left censoring at 0 occurs when a focal patent receives either one or no forward citations, making generality measures either zero or undefined. Patents receiving only one citation also have undefined dispersion. For JTH generality, additional right censoring at 1 occurs when citations are perfectly distributed across technology classes. In contrast, Shannon generality has no upper bound, but remains censored from below at zero.

Tobit estimators are appropriate and consistent in the presence of censoring, provided that the assumptions of normality and homoscedasticity of the error terms hold [89]. Models 5–8 in Table 5 present Tobit estimates, with Models 5 and 7 excluding forward citation controls to establish baseline effects, while Models 6 and 8 include forward citation variables to assess how citation volume affects generality.

**Table 4. AI performance by technological impact (forward citation count).**

| | Model 1 (NB) | Model 2 (NB) | Model 3 (ZINB) | | Model 4 (ZINB) | |
|---|---|---|---|---|---|---|
| | 5-yr FW citations | 5-yr FW w/o self-citations | 5-yr FW citations | Inflate (zero) | 5-yr FW w/o self-citations | Inflate (zero) |
| AI patent | 0.162* | 0.194** | 0.158* | −0.032 | 0.194** | 0.088 |
| | (0.063) | (0.066) | (0.069) | (0.496) | (0.072) | (0.444) |
| Number of claims | 0.020*** | 0.020*** | 0.020*** | −0.015 | 0.020*** | −0.008 |
| | (0.002) | (0.002) | (0.002) | (0.009) | (0.002) | (0.006) |
| Number of inventors | 0.036*** | 0.036*** | 0.036*** | 0.121 | 0.036*** | 0.098 |
| | (0.009) | (0.009) | (0.009) | (0.063) | (0.009) | (0.055) |
| Number of CPCs (ln) | 0.209*** | 0.202*** | 0.214*** | 0.291 | 0.205*** | 0.144 |
| | (0.017) | (0.018) | (0.016) | (0.214) | (0.018) | (0.186) |
| Backward citations (ln) | 0.076*** | 0.087*** | 0.070*** | −0.685*** | 0.083*** | −0.665*** |
| | (0.014) | (0.014) | (0.015) | (0.144) | (0.015) | (0.138) |
| Non-patent references (ln) | −0.037* | −0.050** | −0.039* | 0.068 | −0.052** | 0.094 |
| | (0.018) | (0.016) | (0.019) | (0.132) | (0.017) | (0.110) |
| Days to grant | −0.001*** | −0.001*** | −0.001*** | 0.004*** | −0.001*** | 0.004*** |
| | (0.000) | (0.000) | (0.000) | (0.000) | (0.000) | (0.000) |
| Patent family size (ln) | 0.188*** | 0.145*** | 0.184*** | −0.679*** | 0.142*** | −0.628*** |
| | (0.031) | (0.026) | (0.032) | (0.148) | (0.027) | (0.144) |
| Constant | 0.278** | 0.110 | 0.186 | −8.828*** | −0.001 | −9.308*** |
| | (0.101) | (0.098) | (0.095) | (1.025) | (0.085) | (1.064) |
| lnα | 0.618*** | 0.638*** | | 0.600*** | | 0.619*** |
| | (0.076) | (0.086) | | (0.079) | | (0.089) |
| Log likelihood | −147,428 | −141,914 | −147,299 | | −141747 | |
| Pseudo R$^2$ | 0.0607 | 0.0642 | 0.0618 | | 0.0652 | |

Notes: Robust standard errors clustered by filing year in parentheses. *** $p<0.001$, ** $p<0.01$, * $p<0.05$.

All models include filing year and technology field (WIPO) fixed effects.

Given the strength of Tobit's distributional assumptions, we also apply the Heckman selection model as a robustness check (Models 9–10 in Table 6). In this specification, the outcome equation estimates generality among the patents receiving two or more forward citations (for whom generality is computable), while the selection equation models the likelihood of having computable generality (i.e., excluding patents with zero or one citation). We use art unit workload variables—average days to grant and patent application volume by art unit—as exclusion restrictions in the selection equation. These variables plausibly affect examination thoroughness and citation likelihood but should not directly influence generality conditional on receiving citations.

To test Hypothesis 3, we examine technological complementarity as the dependent variable, which is bounded between 0 and 1 by construction. This bounded nature requires estimation approaches that account for the restricted support of the outcome variable. We employ three specifications to ensure robustness (Models 11–16 in Table 7).

First, we estimate ordinary least squares (OLS) models as a baseline (Models 11–12). While OLS does not explicitly account for the bounded nature of the dependent variable, it provides a straightforward interpretation and serves as a benchmark for comparison.

**Table 5. AI performance by technological generality (JTH & Shannon Index).**

| | Model 5 (Tobit) | Model 6 (Tobit) | Model 7 (Tobit) | Model 8 (Tobit) |
|---|---|---|---|---|
| | 5-yr JTH | 5-yr JTH w/o self-citations | 5-yr Shannon | 5-yr Shannon w/o self-citations |
| AI patent | 0.085** | 0.101*** | 0.037** | 0.039** |
| | (0.028) | (0.027) | (0.013) | (0.011) |
| Forward citations count(log) | | 0.485*** | | 1.177*** |
| | | (0.035) | | (0.072) |
| Dummy for zero forward citations | | −2.006*** | | −4.374*** |
| | | (0.057) | | (0.119) |
| Number of claims | 0.012*** | 0.012*** | 0.005*** | 0.004*** |
| | (0.002) | (0.002) | (0.001) | (0.001) |
| Number of inventors | 0.024*** | 0.023*** | 0.008** | 0.007** |
| | (0.005) | (0.006) | (0.003) | (0.002) |
| Number of CPCs (ln) | 0.122*** | 0.118*** | 0.045*** | 0.039*** |
| | (0.017) | (0.021) | (0.007) | (0.007) |
| Backward citations (ln) | 0.049** | 0.046** | 0.015* | 0.013 |
| | (0.016) | (0.017) | (0.007) | (0.007) |
| Non-patent references (ln) | −0.001 | 0.004 | 0.004 | 0.006 |
| | (0.014) | (0.013) | (0.005) | (0.005) |
| Days to grant | −0.000*** | −0.000*** | −0.000*** | −0.000*** |
| | (0.000) | (0.000) | (0.000) | (0.000) |
| Patent family size (ln) | 0.001 | −0.010 | 0.004 | 0.001 |
| | (0.017) | (0.017) | (0.007) | (0.007) |
| Test statistics: model variance | 0.511*** | 0.211*** | 2.749*** | 1.031*** |
| | (0.051) | (0.014) | (0.247) | (0.059) |
| Constant | −0.720*** | −0.766*** | 0.178* | 0.172* |
| | (0.168) | (0.181) | (0.078) | (0.078) |
| Observations | 10,695 | 10,695 | 10,695 | 10,695 |
| Log likelihood | −4173 | −3905 | −2544 | −2367 |
| Chi² | . | . | 0.140 | 0.146 |

Notes: Robust standard errors clustered by filing year in parentheses. *** p<0.001, ** p<0.01, * p<0.05.

All models include filing year and technology field (WIPO) fixed effects. model variance represents the variance of the error term in Tobit models.

Second, we use two-limit Tobit regression with lower bound at 0 and upper bound at 1 [90] for Models 13–14. The Tobit specification accounts for censoring at both boundaries and is appropriate when the latent complementarity variable may extend beyond the observed [0,1] range but is truncated by the measurement process.

Third, we employ fractional logit regression [91] for Models 15–16, which is specifically designed for dependent variables that are proportions or fractions bounded in the unit interval. The fractional logit model uses a quasi-maximum likelihood estimator and does not require assumptions about the distribution of the dependent variable, making it robust to various data-generating processes. The model is specified as:

$$E\left(\text{Complementarity}_i \mid X_i\right) = \Lambda(\beta^0 + \beta^1 AI_i + \gamma X_i)$$

where $\Lambda(\cdot)$ is the logistic cumulative distribution function. This specification ensures that predicted values remain within [0,1] regardless of the covariate values.

**Table 6. AI performance by technological generality (JTH & Shannon with selection model).**

| | Model 9 (Heckman) 5-yr JTH w/o self-citations | | Model 10 (Heckman) 5-yr Shannon w/o self-citations | |
| --- | --- | --- | --- | --- |
| | Main | Selection | Main | Selection |
| AI patent | 0.020* | 0.058* | 0.052* | 0.058* |
| | (0.010) | (0.027) | (0.023) | (0.027) |
| Forward citations count(log) | 0.118*** | | 0.403*** | |
| | −0.008 | | −0.018 | |
| Number of claims | 0.001 | 0.010*** | 0.003* | 0.010*** |
| | (0.001) | (0.002) | (0.002) | (0.002) |
| Number of inventors | −0.007* | −0.003 | −0.014* | −0.003 |
| | (0.003) | (0.008) | (0.007) | (0.008) |
| Number of CPCs (ln) | 0.006 | | 0.012 | |
| | (0.007) | | (0.017) | |
| Backward citations (ln) | 0.011* | | 0.026* | |
| | (0.005) | | (0.012) | |
| Patent family size (ln) | −0.033*** | | −0.069*** | |
| | (0.007) | | (0.015) | |
| Art unit avg. days | | −0.000*** | | −0.000*** |
| | | (0.000) | | (0.000) |
| Art unit volume (ln) | | 0.198*** | | 0.198*** |
| | | (0.013) | | (0.013) |
| ρ | | 0.122 | | 0.158* |
| | | (0.071) | | (0.075) |
| lnγ | | −1.378*** | | −0.520*** |
| | | (0.015) | | (0.016) |
| Constant | 0.062 | −1.342*** | −0.113 | −1.344*** |
| | (0.032) | (0.140) | (0.076) | (0.140) |
| Observations | 10,695 | 10,695 | 10,695 | 10,695 |
| Log likelihood | −5897 | −5897 | −8256 | −8256 |
| Chi² | 289.2 | 289.2 | 551.2 | 551.2 |

Notes: Standard errors in parentheses. *** p<0.001, ** p<0.01, * p<0.05

Heckman selection models do not include fixed effects due to convergence issues. Selection equation uses Art unit workload variables (Avg. days and volume) as exclusion restrictions.

WIPO technology field fixed effects. Standard errors are clustered by filing year to account for potential correlation in outcomes among patents filed in the same year. The Heckman models (Table 6) do not include fixed effects due to convergence issues, though this limitation is mitigated by the use of exclusion restrictions that identify the selection equation.

## 4. Results

Before testing our hypotheses, we provide an overview of the trends in Samsung Electronics' AI patenting activity.

Between 1987 and 2018, Samsung applied for a total of 6,915 AI-related patents. AI patenting activity began to accelerate in 1996, with applications exceeding 100 per year. This upward trend intensified in the mid-2010s, with annual filings surpassing 500 between 2013 and 2015. Notably, AI patenting grew at a faster rate than overall patenting activity. Prior to 2010, AI patents accounted for approximately 4~8% of Samsung's total filings; this share increased to 9~12% after 2010. These trends are depicted in Fig 1, where AI patents (left axis) and total patents (right axis) are plotted over time.

**Table 7. AI performance by technological complementarity.**

| | Model 11 (OLS) | Model 12 (OLS) | Model 13 (Tobit) | Model 14 (Tobit) | Model 15 (Flogit) | Model 16 (Flogit) |
|---|---|---|---|---|---|---|
| | 5-yr cmp. | 10-yr cmp. | 5-yr cmp. | 10-yr cmp. | 5-yr cmp | 10-yr cmp. |
| AI patent | 0.005* | 0.005* | 0.009* | 0.008* | 0.068* | 0.065* |
| | (0.003) | (0.002) | (0.004) | (0.004) | (0.028) | (0.028) |
| Number of claims | 0.001** | 0.001** | 0.001** | 0.001** | 0.008** | 0.008** |
| | (0.000) | (0.000) | (0.000) | (0.000) | (0.003) | (0.003) |
| Number of inventors | 0.002*** | 0.002*** | 0.004*** | 0.003*** | 0.023*** | 0.023*** |
| | (0.001) | (0.001) | (0.001) | (0.001) | (0.006) | (0.006) |
| Number of CPCs (ln) | 0.043*** | 0.041*** | 0.088*** | 0.084*** | 0.530*** | 0.529*** |
| | (0.002) | (0.002) | (0.004) | (0.004) | (0.027) | (0.028) |
| Backward citations (ln) | −0.005** | −0.004** | −0.007** | −0.006** | −0.051** | −0.049** |
| | (0.002) | (0.001) | (0.002) | (0.002) | (0.016) | (0.015) |
| Non-patent references (ln) | 0.002 | 0.001 | 0.001 | 0.001 | 0.012 | 0.010 |
| | (0.002) | (0.002) | (0.003) | (0.002) | (0.019) | (0.018) |
| Days to grant | −0.000 | −0.000 | 0.000 | 0.000 | −0.000 | −0.000 |
| | (0.000) | (0.000) | (0.000) | (0.000) | (0.000) | (0.000) |
| Patent family size (ln) | −0.006* | −0.005* | −0.011** | −0.010** | −0.071** | −0.068** |
| | (0.003) | (0.002) | (0.004) | (0.003) | (0.026) | (0.026) |
| Test statistics: model variance | | | 0.019*** | 0.017*** | | |
| | | | (0.001) | (0.001) | | |
| Constant | 0.070*** | 0.072*** | −0.052** | −0.041* | −2.953*** | −2.933*** |
| | (0.013) | (0.013) | (0.018) | (0.019) | (0.107) | (0.113) |
| Observations | 10,695 | 10,695 | 10,695 | 10,695 | 10,695 | 10,695 |
| $R^2$ | 0.205 | 0.197 | | | | |
| Log likelihood | | | 456.1 | 805.5 | −3380 | −3257 |
| $Chi^2$ | | | . | . | 4.950e + 10 | 2.700e + 11 |

Notes: Robust standard errors clustered by filing year in parentheses. *** p<0.001, ** p<0.01, * p<0.05.

All models include filing year and technology field (WIPO) fixed effects. Model var. represents the variance of the error term in Tobit models.

We also examine changes in the composition of Samsung's AI patent portfolio by comparing two periods: 2000–2009 and 2010–2018.

As shown in Fig 2, vision-related AI patents consistently represent the largest share, increasing from 32% to 38% between the two periods. Knowledge Processing (kr) patents remained the second-largest category, although their share slightly declined from 24% to 22%. Planning-related patents also declined (18% to 16%), and hardware-related AI patents experienced a slight decrease (15% to 14%). Speech-related AI patents saw a slight increase, from 6% to 7%. Meanwhile, the shares of machine learning (ml), natural language processing (nlp), and evolutionary computation (evo) patents remained small and relatively stable. These shifts suggest an evolving focus in Samsung's AI innovation strategy, with a growing emphasis on vision-related AI applications over time.

### 4.1. Hypothesis 1: Technological impact

Table 4 presents the results for Hypothesis 1, which predicts that AI patents exhibit higher technological impact than comparable non-AI patents. We estimated negative binomial (NB) and zero-inflated negative binomial (ZINB) regression models to account for overdispersion and excess zeros in forward citation counts.

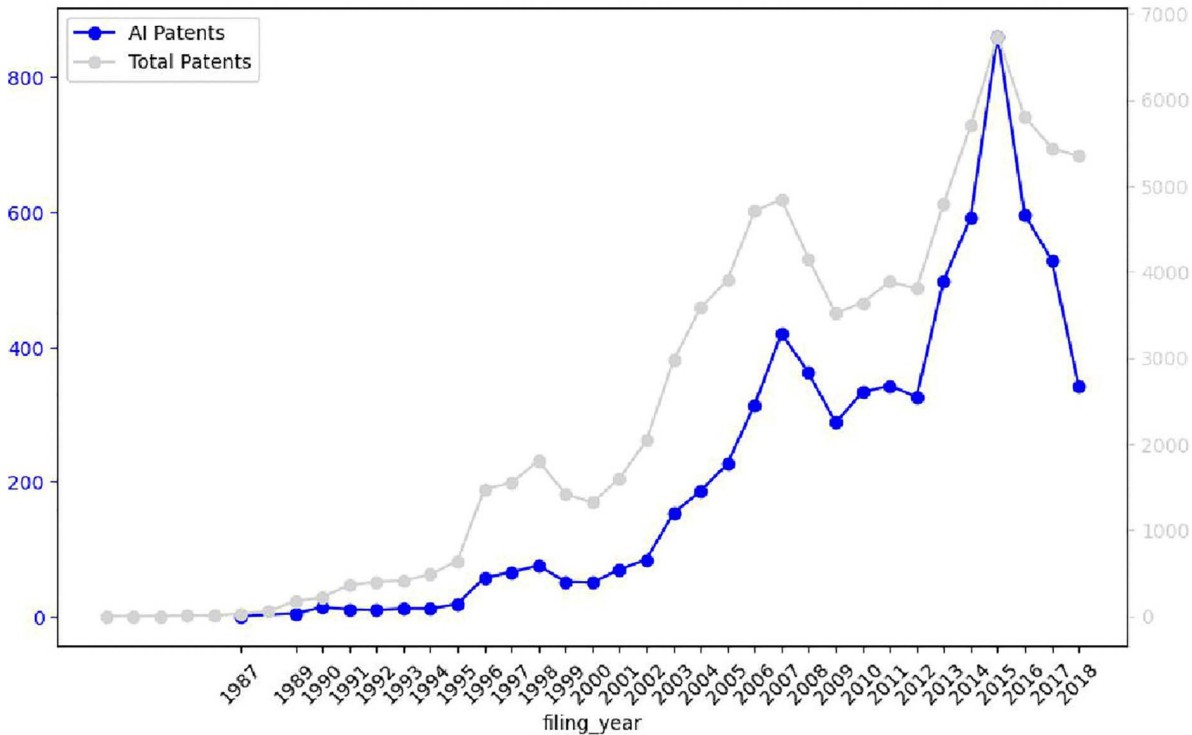

**Fig 1. Annual patent filings of Samsung Electronics (1982-2018).**

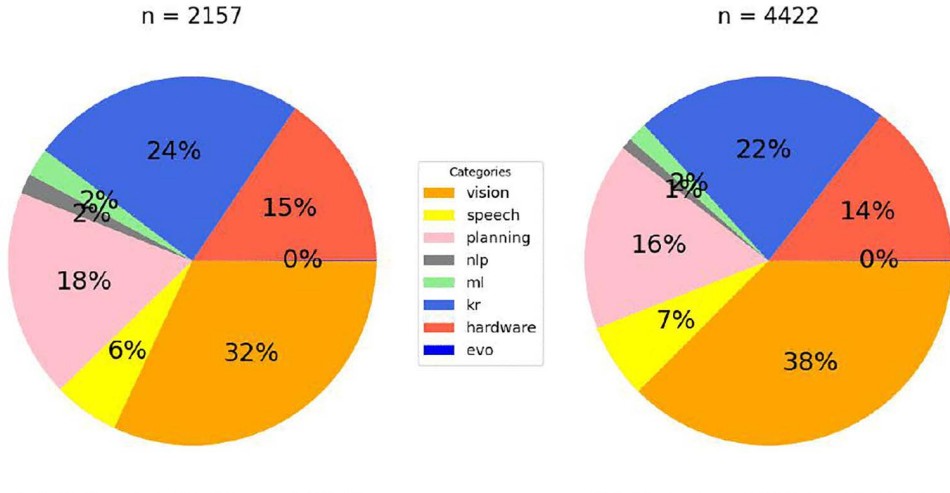

n = 2157    n = 4422

Categories
- vision
- speech
- planning
- nlp
- ml
- kr
- hardware
- evo

(a) Between 2000 and 2009    (b) Between 2010 and 2018

**Fig 2. Breakdown of Samsung's AI patents.**

Across all specifications, AI patents demonstrate significantly higher forward citations than matched non-AI patents. In Model 1 (standard NB), the coefficient for AI patent is 0.162 ($p < 0.05$), indicating that AI patents receive approximately 17.6% ($e^{0.162} \approx 1.176$) more forward citations than comparable non-AI patents. This effect remains robust when excluding self-citations (Model 2: $\beta = 0.194$, $p < 0.01$) and when using ZINB specifications (Model 3: $\beta = 0.158$, $p < 0.05$; Model 4: $\beta = 0.194$, $p < 0.01$).

The dispersion parameter (ln$\alpha$) is statistically significant across all models (ranging from 0.600 to 0.638, all $p < 0.001$), confirming substantial overdispersion and justifying the use of negative binomial models over Poisson regression. The pseudo $R^2$ values range from 0.0607 to 0.0652, indicating moderate model fit. ZINB models show slightly improved fit compared to standard NB models, though the substantive conclusions remain consistent.

Among control variables, technological breadth (number of CPCs: $\beta = 0.205 \sim 0.214$, $p < 0.001$), patent scope (number of claims: $\beta = 0.020$, $p < 0.001$), inventor team size (number of inventors: $\beta = 0.036$, $p < 0.001$), and patent family size ($\beta = 0.142 \sim 0.188$, $p < 0.001$) all positively predict forward citations. Notably, days to grant shows a negative effect ($\beta = -0.001$, $p < 0.001$), suggesting that patents requiring longer examination receive fewer subsequent citations, possibly reflecting lower technological value or greater complexity that limits accessibility.

These findings provide strong support for Hypothesis 1, demonstrating that AI patents generate significantly more downstream technological development than carefully matched non-AI patents, even after controlling for patent characteristics, technological domains, and temporal effects.

## 4.2. Hypothesis 2: Technological generality

Table 5 and 6 present results for Hypothesis 2, which predicts that AI patents exhibit higher technological generality than comparable non-AI patents. We employed Tobit regression models to account for censoring in generality measures (Table 5) and Heckman selection models as robustness checks (Table 6), using both Herfindahl-based (JTH) and Shannon entropy measures of generality.

Tobit regression results (Table 6) provide strong support for Hypothesis 2. For *JTH generality*, AI patents show significantly higher generality scores (Model 5: $\beta = 0.085$, $p < 0.01$; Model 6 w/o self-citations: $\beta = 0.101$, $p < 0.001$). For *Shannon generality*, the effects are similarly positive and significant (Model 7: $\beta = 0.037$, $p < 0.01$; Model 8 w/o self-citations: $\beta = 0.039$, $p < 0.01$). The consistency across both measures—which differ in their sensitivity to citation dispersion—strengthens confidence in these findings.

The inclusion of forward citation controls substantially attenuates AI coefficients, as expected given that generality requires citations to be calculable. Models 5 and 7 exclude forward citation variables, while these controls are inherently captured through the selection of patents with computable generality. Forward citation count (log) shows strong positive associations with generality (Model 5: $\beta = 0.485$, $p < 0.001$; Model 7: $\beta = 1.177$, $p < 0.001$), while the dummy for zero citations shows strong negative effects (Model 5: $\beta = -2.006$, $p < 0.001$; Model 7: $\beta = -4.374$, $p < 0.001$), confirming that patents receiving no citations have zero or undefined generality.

Heckman selection models (Table 6) account for potential selection bias arising from the requirement that patents receive at least two forward citations for generality to be computable. Results remain supportive of Hypothesis 2, though effect sizes are smaller in the outcome equations. For *JTH generality*, AI patents in the selected sample (those receiving 2 + citations) show $\beta = 0.020$ ($p < 0.05$) higher generality. For *Shannon generality*, the effect is $\beta = 0.052$ ($p < 0.05$). The selection equations indicate that AI patents are more likely to receive sufficient citations for generality calculation ($\beta = 0.058$, $p < 0.05$ in both models), with art unit workload variables (average days and volume) serving as valid exclusion restrictions.

The $\gamma$ parameters in Heckman models are small and only marginally significant (Model 10: $\rho = 0.158$, $p < 0.05$), suggesting limited selection bias. Chi-square statistics (289.2 for JTH, 551.2 for Shannon) indicate that the Heckman models provide improved fit over standard Tobit or OLS specifications.

Control variables show consistent patterns across specifications. Technological breadth (number of CPCs), patent scope (claims), and inventor team size positively predict generality, indicating that more complex and collaborative innovations tend to influence diverse technological domains. Days to grant shows a negative effect, consistent with the impact results.

Overall, these findings strongly support Hypothesis 2. AI patents demonstrate higher generality than matched non-AI patents across multiple measures and model specifications, indicating that AI innovations influence subsequent technological development across more diverse domains than comparable non-AI technologies.

### 4.3. Hypothesis 3: Technological complementarity

Table 7 presents results for Hypothesis 3, which predicts that AI patents exhibit higher technological complementarity than comparable non-AI patents. We employed three estimation approaches—OLS, two-limit Tobit, and fractional logit—to account for the bounded [0,1] nature of complementarity scores.

Results provide consistent support for Hypothesis 3 across all specifications. In OLS models (Models 11–12), AI patents show significantly higher complementarity scores for both 5-year ($\beta = 0.005$, $p < 0.05$) and 10-year ($\beta = 0.005$, $p < 0.05$) windows. Two-limit Tobit models (Models 13–14), which account for censoring at both 0 and 1 boundaries, yield similar results (5-year: $\beta = 0.009$, $p < 0.05$; 10-year: $\beta = 0.008$, $p < 0.05$). Fractional logit models (Models 15–16), which use quasi-maximum likelihood estimation specifically designed for fractional dependent variables, show positive effects (5-year: $\beta = 0.068$, $p < 0.05$; 10-year: $\beta = 0.065$, $p < 0.05$). While fractional logit coefficients are not directly comparable to OLS or Tobit due to the logistic transformation, the consistent direction and significance across all three approaches strongly support the hypothesis.

The variance parameters in Tobit models are statistically significant (0.019 for 5-year, 0.017 for 10-year, both $p < 0.001$), indicating substantial unexplained variation in complementarity scores. OLS models achieve $R^2$ values of 0.205 and 0.197, suggesting that our model explains approximately 20% of variation in complementarity. Log likelihood values for Tobit (456.1, 805.5) and fractional logit (−3380, −3257) models indicate adequate fit.

Control variables reveal important patterns. Technological breadth (number of CPCs) shows the strongest positive effect across all models ($\beta = 0.041 \sim 0.088$ in linear models, $\beta = 0.529 \sim 0.530$ in fractional logit, all $p < 0.001$), indicating that patents spanning multiple technology classes exhibit higher complementarity. Team size (number of inventors) also positively predicts complementarity ($\beta = 0.002 \sim 0.004$, $p < 0.001$), suggesting that collaborative innovations create stronger synergistic relationships with other technologies. Interestingly, backward citations show negative effects ($\beta = −0.004$ to −0.007, $p < 0.01$), potentially indicating that patents building extensively on prior art establish weaker novel complementarities. Patent family size also shows negative associations with complementarity ($\beta = −0.005$ to −0.011, $p < 0.01$), suggesting that innovations pursued internationally may reflect standalone value rather than systematic integration with other technologies.

The consistency of findings across 5-year and 10-year windows indicates that AI's complementarity advantage is stable across different temporal scopes for measuring technology co-occurrence patterns. The robustness across OLS, Tobit, and fractional logit specifications—each making different distributional assumptions—strengthens confidence in these findings.

These results provide strong support for Hypothesis 3, demonstrating that AI patents exhibit higher technological complementarity than matched non-AI patents. AI innovations co-occur with diverse technologies more frequently than random expectations would predict, exhibiting systematic, non-random patterns consistent with complementarity rather than coincidental overlap.

Overall, our empirical analyses provide strong and consistent support for all three hypotheses. AI patents demonstrate significantly higher technological impact (H1), broader generality (H2), and stronger complementarity (H3) compared to carefully matched non-AI patents. These findings are robust across multiple model specifications, alternative measures,

and the inclusion of comprehensive controls for patent characteristics, technological domains, and temporal effects. The consistent patterns across all three dimensions suggest that AI innovations exhibit distinctive technological characteristics consistent with broadly applicable, transformative technologies.

## 4.4. Robustness checks

To assess the sensitivity of our findings to matching specifications, we implemented three alternative CEM approaches alongside our main specification.

Figs 3 and 4 presents the sample sizes and covariate balance achieved across the four matching approaches, respectively. The original CEM specification (N = 10,695) achieves the strongest covariate balance, with standardized differences below 0.1 for all matching variables—the conventional threshold for adequate balance. The 1:1 matching approach (N = 9,244) requires exact one-to-one correspondence between AI and non-AI patents, yielding a smaller but balanced sample. Same-year matching (N = 8,708) imposes stricter temporal alignment by requiring exact filing year matches rather than ±1 year windows. The no-CPC specification (N = 19,284) removes the technology classification overlap requirement, substantially increasing sample size but resulting in larger standardized differences, particularly for the number of CPC codes (0.244), suggesting greater technological heterogeneity in this sample.

We re-estimated all models across these three alternative specifications. Results demonstrate substantial robustness across specifications. For technological impact (Table 8), AI patents maintain significantly higher forward citations in both 1:1 matching (β = 0.164 ~ 0.186, p < 0.05) and same-year matching (β = 0.187 ~ 0.215, p < 0.01 ~ 0.05) specifications. In the no-CPC sample, coefficients are positive but not statistically significant (β = 0.099 ~ 0.125, p > 0.05), likely reflecting the

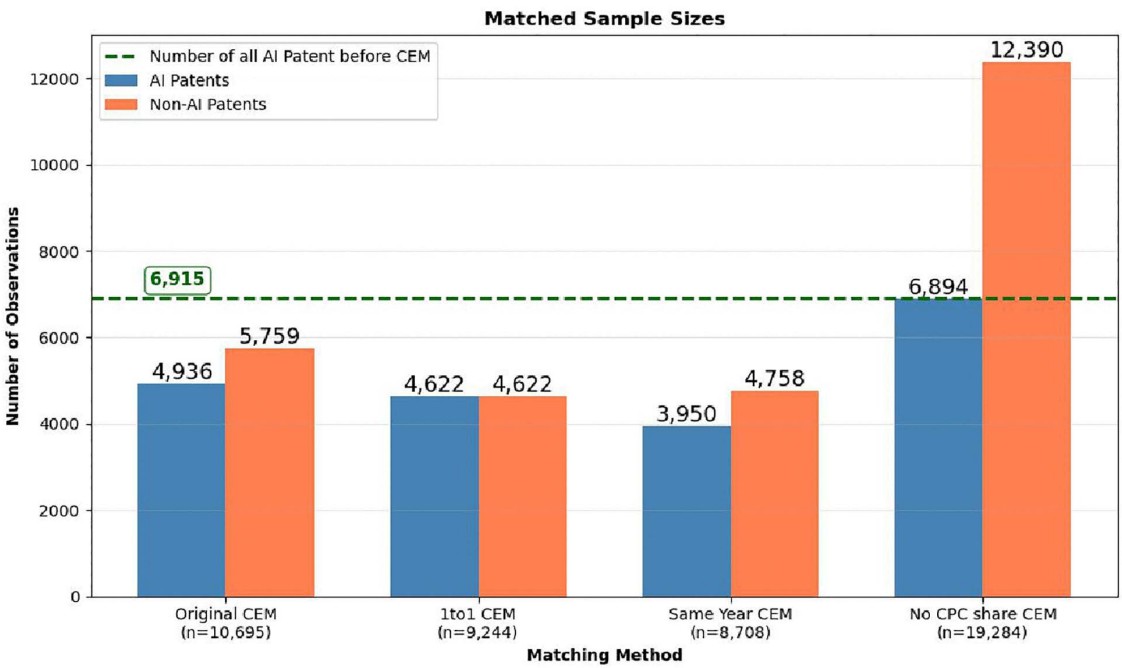

Note: The figure displays the number of AI patents (blue) and matched non-AI patents (orange) across four CEM specifications. The original CEM (N=10,695) balances sample size with covariate balance. The 1:1 CEM (N=9,244) enforces exact one-to-one matching. Same-year CEM (N=8,708) requires exact filing year matches. No-CPC CEM (N=19,284) removes technology classification overlap requirements, yielding the largest sample.

**Fig 3. Coarsened exact matching sample composition.**

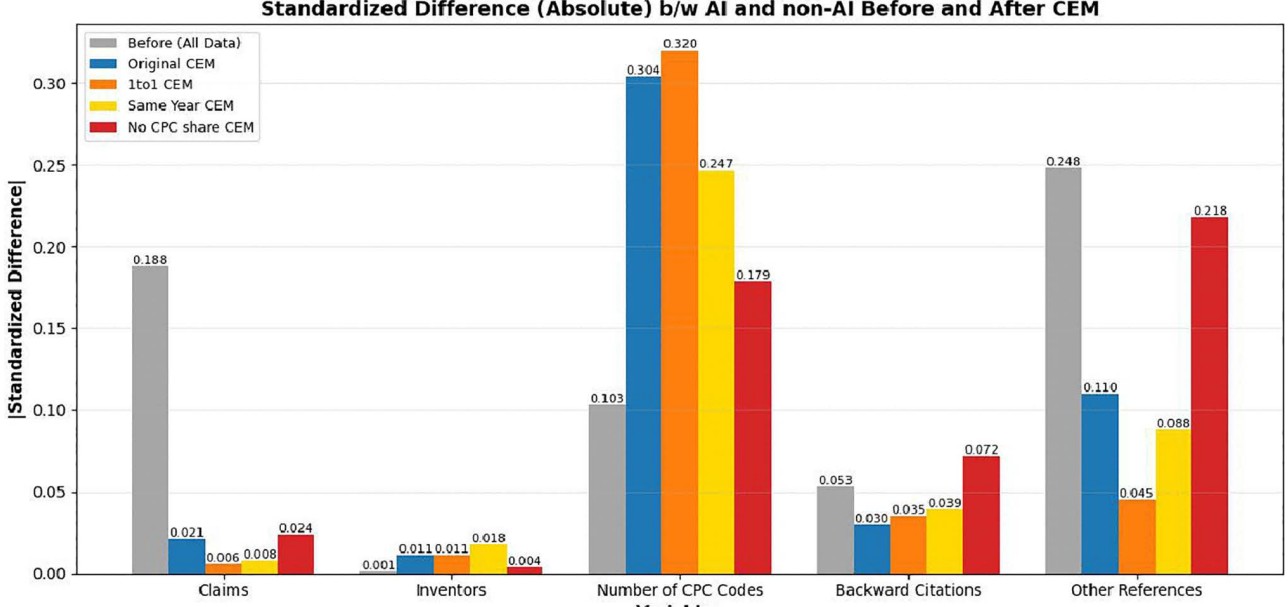

Note: Standardized differences (absolute values) between AI and non-AI patents before matching (gray) and after matching using four CEM specifications. Values below 0.1 (dashed line) indicate adequate balance. Original CEM achieves the strongest balance across all matching variables. No-CPC specification shows larger imbalance, particularly for number of CPC codes (0.244), reflecting greater technological heterogeneity when classification overlap is not required.

**Fig 4. Covariate balance across CEM specifications.**

increased technological heterogeneity, suggesting that comparing patents within similar technological domains is important for isolating AI's distinctive impact.

For generality (Tables 9 through 10), AI coefficients remain positive and significant across all three alternative matching specifications for both JTH and Shannon measures, with effect sizes comparable to or larger than main results. Notably, generality effects are robust even in the no-CPC sample as shown in Table 11 (β = 0.074 ~ 0.089 for JTH, β = 0.183 ~ 0.221 for Shannon, all p < 0.01), indicating that AI's broader technological influence persists regardless of technology classification overlap.

For complementarity (Table 12), AI coefficients are positive and significant across all specifications and all three model types (OLS, Tobit, fractional logit), including the no-CPC sample (β = 0.005 ~ 0.010, p < 0.05–0.01). The consistency of generality and complementarity findings across all matching specifications—including when technology classification overlap is not required—strengthens confidence that these effects capture fundamental characteristics of AI innovations rather than artifacts of matching procedures. Overall, these robustness checks confirm that our main findings are not driven by specific matching decisions and remain stable across alternative sample construction approaches.

We conducted additional robustness checks to address potential concerns about field and temporal heterogeneity in citation patterns, employing two alternative normalization approaches: (1) percentile ranks within filing-year × WIPO technology field combinations, and (2) z-scores within time periods.

$$Z_{score}\ Forward\ Citation_{it} = \frac{FC_{it} - mean(FC_t)}{sd(FC_t)}$$

**Table 8. AI performance by technological impact (NB using three alternative CEM).**

| | Model 17 | Model 18 | Model 19 | Model 20 | Model 21 | Model 22 |
|---|---|---|---|---|---|---|
| | 1:1 CEM | | Same-year CEM | | No CPC CEM | |
| | 5-yr FW citations | w/o self-citations | 5-yr FW citations | w/o self-citations | 5-yr FW citations | w/o self-citations |
| AI patent | 0.164* | 0.186* | 0.187* | 0.215** | 0.099 | 0.125 |
| | (0.074) | (0.074) | (0.079) | (0.079) | (0.091) | (0.093) |
| Number of claims | 0.023*** | 0.025*** | 0.025*** | 0.026*** | 0.024*** | 0.023*** |
| | (0.004) | (0.004) | (0.003) | (0.004) | (0.003) | (0.004) |
| Number of inventors | 0.084*** | 0.079*** | 0.079*** | 0.075** | 0.046** | 0.045* |
| | (0.018) | (0.021) | (0.021) | (0.023) | (0.017) | (0.019) |
| Number of CPCs | 0.247*** | 0.253*** | 0.266*** | 0.288*** | 0.230*** | 0.234*** |
| | (0.029) | (0.033) | (0.039) | (0.044) | (0.031) | (0.033) |
| Backward citations | 0.075* | 0.074* | 0.062 | 0.058 | 0.066* | 0.066 |
| | (0.030) | (0.032) | (0.032) | (0.033) | (0.030) | (0.034) |
| NP references | −0.009 | −0.005 | −0.026 | −0.018 | −0.022 | −0.023 |
| | (0.035) | (0.035) | (0.036) | (0.040) | (0.025) | (0.026) |
| Days to grant | −0.001*** | −0.001*** | −0.001*** | −0.001*** | −0.001*** | −0.001*** |
| | (0.000) | (0.000) | (0.000) | (0.000) | (0.000) | (0.000) |
| Patent family size | 0.114** | 0.061 | 0.119*** | 0.071* | 0.128* | 0.092 |
| | (0.037) | (0.040) | (0.034) | (0.035) | (0.050) | (0.047) |
| Constant | 0.293 | 0.169 | 0.196 | 0.020 | 0.348* | 0.274 |
| | (0.267) | (0.272) | (0.299) | (0.319) | (0.144) | (0.183) |
| lnα | 0.739*** | 0.812*** | 0.742*** | 0.814*** | 0.784*** | 0.847*** |
| | (0.082) | (0.096) | (0.077) | (0.087) | (0.077) | (0.092) |
| Observations | 9,244 | 9,244 | 8,708 | 8,708 | 19,284 | 19,284 |
| Pseudo $R^2$ | 0.0684 | 0.0695 | 0.0661 | 0.0676 | 0.0552 | 0.0563 |
| Log likelihood | −13986 | −13315 | −13017 | −12355 | −31263 | −29909 |

Notes: Robust standard errors clustered by filing year in parentheses. *** $p < 0.001$, ** $p < 0.01$, * $p < 0.05$.

All models include filing year and technology field (WIPO) fixed effects.

where mean($FC_t$) is the mean of forward citations of patents filed in the same year and sd($FC_t$) is the standard deviation of forward citations of patents filed in the same year.

Results presented in Appendix IV in S1 Dataset, Table A3 demonstrate that our main findings remain substantively unchanged across both normalization methods, confirming that the observed effects are robust to field-specific and time-varying citation practices.

In additional sensitivity analyses, we examined 3-year forward citation windows for impact and generality measures (Appendix V, Table A4 in S1 Dataset). Results exhibit similar directional patterns but with attenuated effect sizes and reduced statistical significance relative to 5-year windows, consistent with prior research showing that shorter windows capture incomplete citation trajectories [67]. Following standard practice in patent research, we retain 5-year windows as our primary specification, which provides more stable measures of technological influence.

### 4.5. Supplementary analysis: AI technology categories

To address potential heterogeneity in the transformative characteristics of AI technologies, we conduct supplementary analyses examining variation across different AI subfields. Tables 13 and 14 present these results, comparing AI technology categories against non-AI patents and examining within-AI heterogeneity, respectively.

**Table 9. AI performance by technological generality (1:1 CEM).**

| | Model 23<br>Tobit | Model 24<br>Tobit | Model 25<br>Tobit | Model 26<br>Tobit |
|---|---|---|---|---|
| | 5-yr JTH | 5-yr JTH<br>w/o self-citations | 5-yr Shannon | 5-yr Shannon<br>w/o self-citations |
| AI patent | 0.104** | 0.113*** | 0.251*** | 0.273*** |
| | (0.032) | (0.034) | (0.075) | (0.079) |
| Forward citations count | 0.013*** | 0.012*** | 0.030*** | 0.029*** |
| | (0.003) | (0.003) | (0.006) | (0.006) |
| Dummy for zero FW | 0.030*** | 0.027*** | 0.072*** | 0.065*** |
| | (0.004) | (0.006) | (0.010) | (0.014) |
| Number of claims | 0.117*** | 0.117*** | 0.275*** | 0.274*** |
| | (0.017) | (0.020) | (0.038) | (0.047) |
| Number of inventors | 0.044** | 0.043* | 0.104** | 0.100* |
| | (0.015) | (0.018) | (0.037) | (0.043) |
| Number of CPCs | 0.002 | 0.006 | 0.009 | 0.017 |
| | (0.019) | (0.018) | (0.043) | (0.042) |
| Backward citations | −0.000*** | −0.000*** | −0.001*** | −0.001*** |
| | (0.000) | (0.000) | (0.000) | (0.000) |
| Non-patent references | −0.025 | −0.035 | −0.052 | −0.076 |
| | (0.020) | (0.021) | (0.048) | (0.051) |
| Constant | −0.732*** | −0.744*** | −1.710*** | −1.744*** |
| | (0.171) | (0.184) | (0.410) | (0.440) |
| Model variance | 0.514*** | 0.536*** | 2.777*** | 2.914*** |
| | (0.052) | (0.061) | (0.255) | (0.307) |
| Observations | 9,244 | 9,244 | 9,244 | 9,244 |
| Log likelihood | −3483 | −3265 | −4590 | −4281 |

**4.5.1 AI categories compared to non-AI patents.** Table 13 examines how different AI technology categories perform relative to non-AI patents across three outcome measures: forward citations (Model 44), Shannon generality (Model 45), and technological complementarity (Model 46). The coefficients indicate differences for each AI category relative to non-AI patents which is set as base.

We observe substantial heterogeneity across AI subfields. For technological impact (Model 44), speech-related AI patents demonstrate the strongest citation advantage (0.509, $p < 0.01$), followed by hardware (0.227, $p < 0.05$) and knowledge processing (0.160, $p < 0.05$). Notably, NLP patents exhibit negative effects (−0.446, $p < 0.01$), possibly reflecting specialized early applications or classification limitations.

For technological generality (Model 45), machine learning (0.155, $p < 0.05$), vision (0.053, $p < 0.001$), and speech (0.071, $p < 0.05$) patents show significant positive effects compared to non-AI patents, while evolutionary computation patents exhibit negative effects (−0.138, $p < 0.05$).

The complementarity measure (Model 46) reveals that vision (0.031, $p < 0.001$) and machine learning (0.035, $p < 0.01$) patents demonstrate the strongest positive effects, whereas hardware patents show negative effects (−0.035, $p < 0.001$), indicating specialized integration patterns despite their high citation impact.

**4.5.2 Domain-specific generality effects within AI technologies.** Table 14 examines Shannon generality heterogeneity within AI patents only, using vision as the baseline. The analysis distinguishes between "total generality" (without citation controls) and "pure generality" (controlling for forward citations) to separate inherent technological

**Table 10. AI performance by technological generality (Same-year CEM).**

| | Model 27 | Model 28 | Model 29 | Model 30 |
|---|---|---|---|---|
| | 5-yr JTH | 5-yr JTH w/o self-citations | 5-yr Shannon | 5-yr Shannon w/o self-citations |
| AI patent | 0.068* | 0.082** | 0.171** | 0.205** |
| | (0.027) | (0.029) | (0.062) | (0.065) |
| Forward citations count | 0.014*** | 0.013*** | 0.032*** | 0.030*** |
| | (0.002) | (0.002) | (0.005) | (0.005) |
| Dummy for zero FW | 0.032*** | 0.030*** | 0.074*** | 0.071*** |
| | (0.007) | (0.006) | (0.016) | (0.015) |
| Number of claims | 0.116*** | 0.117*** | 0.274*** | 0.275*** |
| | (0.024) | (0.030) | (0.054) | (0.067) |
| Number of inventors | 0.028 | 0.024 | 0.065 | 0.056 |
| | (0.018) | (0.019) | (0.042) | (0.046) |
| Number of CPCs | −0.002 | 0.005 | −0.000 | 0.016 |
| | (0.015) | (0.016) | (0.036) | (0.037) |
| Backward citations | −0.000*** | −0.000*** | −0.001*** | −0.001*** |
| | (0.000) | (0.000) | (0.000) | (0.000) |
| Non-patent references | 0.002 | −0.007 | 0.010 | −0.010 |
| | (0.020) | (0.022) | (0.048) | (0.051) |
| Constant | −0.753*** | −0.818*** | −1.746*** | −1.894*** |
| | (0.182) | (0.200) | (0.422) | (0.465) |
| Model variance | 0.529*** | 0.556*** | 2.794*** | 2.943*** |
| | (0.050) | (0.059) | (0.241) | (0.288) |
| Observations | 8,708 | 8,708 | 8,708 | 8,708 |
| Log likelihood | −3297 | −3068 | −4306 | −3982 |

breadth from citation-mediated effects. The coefficients represent average marginal effects (AME), indicating the average difference for each AI category relative to vision category which is set as base.

The systematic comparison of total versus pure generality reveals three distinct patterns. First, evolutionary computation, hardware, and knowledge processing patents demonstrate persistent negative generality effects that remain significant after controlling for citations. Evolutionary computation patents show the strongest pattern (total: $\beta = -0.232$, $p < 0.001$; pure: $\beta = -0.307$, $p < 0.01$), indicating genuine technological specialization independent of citation impact. Second, planning patents show citation-mediated effects: significantly negative total generality ($\beta = -0.031$, $p < 0.01$) becomes non-significant when controlling for citations (pure: $\beta = -0.017$, not significant), suggesting their lower generality operates primarily through citation pathways. Third, machine learning, NLP, and speech patents demonstrate generality comparable to vision across both specifications, with no significant differences detected.

**4.5.3 Implication of supplementary analysis.** These findings reveal that AI technologies are heterogeneous rather than monolithic. Technological impact and generality do not necessarily coincide: speech patents demonstrate both high citations and broad generality, while hardware and evolutionary computation patents show high impact but inherent specialization. Vision and machine learning emerge as the subfields with the broadest technological reach, joined by speech and NLP in demonstrating comparable cross-domain applicability. The distinction between total and pure generality clarifies that some specialization effects (evolutionary computation, hardware, knowledge processing) represent genuine technological constraints independent of innovation impact, while others (planning) operate primarily through

**Table 11. AI performance by technological generality (No CPC CEM).**

| | Model 31 | Model 32 | Model 33 | Model 34 |
| --- | --- | --- | --- | --- |
| | 5-yr JTH | 5-yr JTH w/o self-citations | 5-yr Shannon | 5-yr Shannon w/o self-citations |
| AI patent | 0.074** | 0.089*** | 0.183** | 0.221*** |
| | (0.026) | (0.026) | (0.063) | (0.062) |
| Forward citations count | 0.011*** | 0.010*** | 0.025*** | 0.025*** |
| | (0.002) | (0.002) | (0.003) | (0.003) |
| Dummy for zero FW | 0.016*** | 0.016*** | 0.039*** | 0.037*** |
| | (0.004) | (0.004) | (0.009) | (0.009) |
| Number of claims | 0.109*** | 0.110*** | 0.263*** | 0.264*** |
| | (0.012) | (0.011) | (0.027) | (0.025) |
| Number of inventors | 0.023*** | 0.026*** | 0.060*** | 0.064*** |
| | (0.007) | (0.007) | (0.017) | (0.018) |
| Number of CPCs | −0.007 | −0.003 | −0.016 | −0.007 |
| | (0.010) | (0.011) | (0.023) | (0.025) |
| Backward citations | −0.000*** | −0.000*** | −0.001*** | −0.001*** |
| | (0.000) | (0.000) | (0.000) | (0.000) |
| Non-patent references | −0.012 | −0.015 | −0.021 | −0.031 |
| | (0.013) | (0.014) | (0.032) | (0.034) |
| Constant | −0.643*** | −0.734*** | −1.539*** | −1.744*** |
| | (0.076) | (0.090) | (0.180) | (0.209) |
| Model variance | 0.497*** | 0.512*** | 2.718*** | 2.794*** |
| | (0.042) | (0.049) | (0.204) | (0.239) |
| Observations | 19,284 | 19,284 | 19,284 | 19,284 |
| Log likelihood | −8200 | −7716 | −10905 | −10205 |

Notes: Robust standard errors clustered by filing year in parentheses. *** p<0.001, ** p<0.01, * p<0.05.

All models include filing year and technology field (WIPO) fixed effects.

citation mechanisms. These patterns underscore the importance of disaggregated analysis, suggesting that transformative impact manifests differently across AI subfields—some generating concentrated innovation within narrow domains while others demonstrate broader cross-domain influence.

## 5. Discussion and conclusion

### 5.1. Summary of findings

While AI technologies have attracted substantial attention from business managers and technologists, a systematic understanding of their nature and broader technological impacts remains essential for designing effective strategies and policies [92]. Despite this importance, empirical evidence on the technological characteristics and systemic implications of AI remains limited and fragmented [15,25,77]. In this study, we address this gap by examining the "GPTness" of AI patents, using Samsung Electronics as a representative case.

The GPT literature identifies three defining characteristics of GPTs: continuous technological improvement, pervasiveness across a wide range of uses, and strong technological complementarities with existing and future technologies [10,14,28]. Given the well-documented rapid advancement of AI technologies, this study focuses on the latter two dimensions—namely, whether AI patents are used more broadly across subsequent technological developments

**Table 12. AI performance by technological complementarity (Using three alternative CEM).**

| | Model 35 | Model 36 | Model 37 | Model 38 | Model 39 | Model 40 | Model 41 | Model 42 | Model 43 |
|---|---|---|---|---|---|---|---|---|---|
| | 1:1 CEM | | | Same-year CEM | | | No CPC CEM | | |
| | (OLS) | (Tobit) | (Flogit) | (OLS) | (Tobit) | (Flogit) | (OLS) | (Tobit) | (Flogit) |
| AI patent | 0.007** | 0.010*** | 0.085*** | 0.005** | 0.008*** | 0.062*** | 0.005* | 0.010** | 0.064** |
| | (0.002) | (0.003) | (0.022) | (0.002) | (0.002) | (0.016) | (0.002) | (0.003) | (0.025) |
| Number of claims | 0.001** | 0.001** | 0.009** | 0.001*** | 0.002*** | 0.010*** | 0.000 | 0.000 | 0.001 |
| | (0.000) | (0.000) | (0.003) | (0.000) | (0.000) | (0.002) | (0.000) | (0.000) | (0.002) |
| Number of inventors | 0.002*** | 0.003*** | 0.020*** | 0.003*** | 0.004*** | 0.026*** | 0.003*** | 0.004*** | 0.031*** |
| | (0.000) | (0.001) | (0.005) | (0.001) | (0.001) | (0.007) | (0.000) | (0.001) | (0.005) |
| Number of CPCs (ln) | 0.044*** | 0.089*** | 0.548*** | 0.042*** | 0.086*** | 0.506*** | 0.042*** | 0.101*** | 0.592*** |
| | (0.002) | (0.004) | (0.030) | (0.002) | (0.005) | (0.036) | (0.001) | (0.006) | (0.038) |
| Backward citations (ln) | −0.004** | −0.006** | −0.049*** | −0.005* | −0.006 | −0.049* | −0.005*** | −0.008*** | −0.064*** |
| | (0.001) | (0.002) | (0.015) | (0.002) | (0.003) | (0.020) | (0.001) | (0.002) | (0.010) |
| Non-patent references (ln) | −0.000 | −0.001 | −0.006 | 0.001 | −0.000 | 0.002 | 0.004* | 0.004 | 0.037* |
| | (0.001) | (0.002) | (0.015) | (0.001) | (0.002) | (0.011) | (0.001) | (0.002) | (0.016) |
| Days to grant | 0.000 | 0.000 | 0.000 | −0.000 | 0.000 | −0.000 | −0.000 | −0.000 | −0.000 |
| | (0.000) | (0.000) | (0.000) | (0.000) | (0.000) | (0.000) | (0.000) | (0.000) | (0.000) |
| Patent family size (ln) | −0.007** | −0.014*** | −0.087** | −0.005 | −0.011** | −0.067* | −0.005** | −0.010*** | −0.064*** |
| | (0.002) | (0.004) | (0.027) | (0.003) | (0.004) | (0.029) | (0.001) | (0.002) | (0.016) |
| Constant | 0.065** | −0.059* | −3.038*** | 0.103** | −0.018 | −2.774*** | 0.086*** | −0.034** | −2.706*** |
| | (0.020) | (0.025) | (0.162) | (0.028) | (0.031) | (0.172) | (0.009) | (0.012) | (0.080) |
| Model variance | | 0.018*** | | | 0.019*** | | | 0.023*** | |
| | | (0.001) | | | (0.001) | | | (0.001) | |
| Observations | 9,244 | 9,244 | 9,244 | 8,708 | 8,708 | 8,708 | 19,284 | 19,284 | 19,284 |
| R² | 0.204 | | | 0.214 | | | 0.204 | | |
| Log likelihood | | 566.3 | −2907 | | 536.8 | −2818 | | −1151 | −5379 |

Notes: Robust standard errors clustered by filing year in parentheses. *** p<0.001, ** p<0.01, * p<0.05.

All models include filing year and technology field (WIPO) fixed effects. Model var. represents the variance of the error term in Tobit models.

(pervasiveness) and whether they exhibit stronger technological complementarities with other technologies than non-AI patents (complementarity).

Our empirical analyses provide strong and consistent evidence that AI patents exhibit substantially higher GPTness than non-AI patents. After controlling for a wide range of covariates using multiple estimation strategies and matching designs, we find that AI patents demonstrate significantly higher technological impact, as measured by forward citations (H1), as well as broader technological generality, assessed using both Herfindahl-based and Shannon entropy measures (H2). Together, these findings indicate that AI patents are more pervasive in their downstream technological use than their non-AI counterparts. In addition, we find that AI patents exhibit stronger technological complementarity—captured by systematic co-occurrence patterns with other technologies (H3)—relative to non-AI patents in our sample.

The consistency of results across all three dimensions is particularly noteworthy. While prior research has documented high citation rates for certain AI technologies, our analysis demonstrates that AI's distinctive characteristics extend beyond citation-based impact to encompass both the breadth of technological influence and systematic patterns of technological integration. By triangulating evidence from impact, generality, and complementarity measures, we provide converging support for AI's role as a foundational, GPT-like technology within Samsung's innovation portfolio.

**Table 13. Heterogeneous Effects of AI Categories on Patent Performance Measures.**

| | Model 44<br>5-yr FW<br>w/o self-citations | Model 45<br>5-yr Shannon<br>w/o self-citations | Model 46<br>5-yr<br>Complementarity |
|---|---|---|---|
| Evolutionary computation | 0.386 | −0.138* | −0.125 |
| | (0.356) | (0.058) | (0.078) |
| Hardware | 0.227* | 0.022 | −0.035*** |
| | (0.109) | (0.021) | (0.006) |
| Knowledge processing | 0.160* | 0.025 | 0.006 |
| | (0.068) | (0.014) | (0.008) |
| Machine learning | 0.286 | 0.155* | 0.035** |
| | (0.194) | (0.070) | (0.011) |
| NLP | −0.446** | 0.004 | −0.019 |
| | (0.171) | (0.048) | (0.018) |
| Planning | 0.323 | 0.028 | 0.006 |
| | (0.255) | (0.017) | (0.005) |
| Speech | 0.509** | 0.071* | 0.000 |
| | (0.173) | (0.030) | (0.009) |
| Vision | 0.137 | 0.053*** | 0.031*** |
| | (0.080) | (0.012) | (0.006) |
| lnα | 0.803*** | | |
| | (0.084) | | |
| Constant | 0.185 | 0.176* | −0.047** |
| | (0.214) | (0.078) | (0.018) |
| Observations | 10,695 | 10,695 | 10,695 |
| $R^2$ | | 0.148 | |
| Log likelihood | −15470 | | 508.9 |

Notes: Robust standard errors clustered by filing year in parentheses. *** $p < 0.001$, ** $p < 0.01$, * $p < 0.05$.

All models include filing year and technology field (WIPO) fixed effects. Controls include number of claims, inventors, CPCs (ln), backward citations (ln), other references (ln), days to grant, and patent family size (ln).

Coefficients represent the effect relative to Non-AI patents (reference category).

One potential concern is what underlies the observed differences in generality between AI and non-AI patents—specifically, whether the higher generality of AI patents reflects genuine technological breadth or merely results from their receiving more citations. We argue that the increased generality of AI patents is driven by their inherent cross-domain applicability, rather than by citation volume alone.

First, the coefficient on the AI indicator remains positive and statistically significant in the Tobit models even after controlling for the number of forward citations (Models 5–8 in Table 5). This indicates that AI patents exhibit broader technological influence conditional on citation intensity. Second, the Heckman selection models (Table 6), which explicitly separate the likelihood of receiving a sufficient number of citations (selection stage) from generality conditional on citations (outcome stage), show positive effects of AI patents in both stages. This finding suggests that the generality advantage of AI patents is not driven solely by selection into the cited sample. Third, supplementary analyses comparing total generality (without citation controls) and pure generality (controlling for forward citations) in Table 14 demonstrate that AI subfields such as vision, machine learning, and speech retain levels of generality comparable to the baseline specifications, even after accounting for citation intensity.

**Table 14. Domain-specific effects on technological generality.**

| | Model 47<br>Total generality | Model 48<br>Pure generality |
|---|---|---|
| Evolutionary computation | −0.232*** | −0.307** |
| | [-0.365,-0.099] | [-0.545,-0.069] |
| Hardware | −0.054*** | −0.077*** |
| | [-0.089,-0.019] | [-0.114,-0.040] |
| Knowledge processing | −0.031** | −0.023** |
| | [-0.058,-0.005] | [-0.046,-0.001] |
| Machine Learning | 0.092 | 0.053 |
| | [-0.054,0.238] | [-0.050,0.157] |
| NLP | −0.052 | 0.024 |
| | [-0.145,0.040] | [-0.048,0.097] |
| Planning | −0.031** | −0.017 |
| | [-0.060,-0.002] | [-0.044,0.010] |
| Speech | 0.023 | −0.017 |
| | [-0.043,0.090] | [-0.050,0.017] |

Notes: Standard errors clustered by filing year. * $p < 0.10$, ** $p < 0.05$, *** $p < 0.0$, 95% confidence intervals in brackets.

Baseline: Vision domain. Model 20 estimates total Shannon generality without impact controls.

Model 21 controls for forward citations to isolate pure Shannon generality effects.

Coefficients show marginal effects relative to Vision. 95% confidence intervals in brackets.

Taken together, these results indicate that the higher generality scores observed for AI patents reflect substantive technological breadth across multiple domains, rather than merely increased citation accumulation.

## 5.2. Scholarly contributions

Our findings make the following contributions to advancing understanding of AI innovation and its role in firm-level technological development.

First, we provide firm-level empirical evidence that AI exhibits characteristics consistent with those of general-purpose technologies (GPTs). GPT theory highlights technologies' broad applicability, potential for ongoing improvement, and ability to generate complementary innovations [10]. While recent studies have begun to validate these characteristics at the macro level—through analyses of citation diversity and co-classification patterns [15,16]—firm-level evidence remains scarce. Notably, recent firm-level AI research has primarily focused on AI's impact on corporate outcomes, such as environmental performance [93], ESG ratings [94], and innovation performance [95], treating AI as an input variable rather than examining the structural characteristics of AI technology itself. Our study addresses this gap by analyzing whether AI patents within a focal firm exhibit GPT-like properties through impact, generality, and complementarity measures.

Recent patent-based research has examined organizational antecedents of AI innovation, such as how firm structure and cross-functional collaboration shape AI patenting outcomes [25]. However, these studies focus on what drives AI patent generation rather than on the intrinsic technological characteristics of AI patents themselves. Similarly, Yavuz and Çalik [96] analyzed lagged effects of AI/ML patent intensity on firm performance, treating AI patents as inputs to financial outcomes. Our study complements this stream by examining the technological structure of AI patents—specifically their impact, generality, and complementarity—rather than their downstream performance effects.

Specifically, we introduce a complementarity measure that captures the systematic co-occurrence of AI with other technologies beyond random expectations. This provides novel evidence of systematic, rather than incidental, co-occurrence patterns —advancing GPT theory by demonstrating that the theoretical notion of pervasiveness can be empirically observed through structured technological complementarities at the firm level. In addition, our generality results show that AI patents influence a broad array of downstream innovations across diverse technology domains, not just within narrowly defined technical areas.

Taken together, these findings suggest that the GPT-like nature of AI may accelerate both the rate and breadth of innovation—within AI itself and across unrelated domains—thereby generating amplified economic impacts, in line with the predictions of Schumpeterian growth theory [11–13].

Second, we contribute methodologically to innovation studies by demonstrating how multiple patent-based indicators—impact, generality, and complementarity—can be combined to assess the characteristics of transformative technologies. While prior research has typically focused on single-dimension measures (e.g., forward citations), our multi-dimensional framework offers a more holistic approach to evaluating technological influence. In particular, the complementarity metric represents a novel application of co-occurrence analysis to detect systematic technology integration. This framework can be applied to study other potentially foundational technologies and to trace the evolution of GPT-like features across time and firms.

## 5.3. Practical implications

Our results have implications to various stakeholders in AI innovation and implementation.

To managers, our findings indicate the strategic importance of building internal AI capabilities, even in applications that might not provide immediate commercial benefits. The high generality and complementarity of AI patents observed suggest that AI capabilities can facilitate innovations in a variety of technological areas in the firm. Our heterogeneity analysis shows that the various types of AI have different strengths, including the high complementarity of Vision AI and the high generality of Machine Learning, which means that managers would be better off diversifying their AI portfolio instead of focusing their resources on one AI area. The especially high complementarity effects indicate that the value of AI is not only in its direct uses but also in the fact that it allows integration with other technologies. Managers must thus not think of AI as an isolated innovation to be isolated in specialized units, but as a strategic enabler to be systematically incorporated in the wider innovation portfolio.

To employees, these results indicate that AI-related skills can provide sustainable career value. Recent experimental data show that AI implementation in the workplace can increase the perceived job autonomy of employees instead of reducing it, especially when employees have time to adjust to AI-supported workflows [97]. The generality of AI technologies is high, which means that the knowledge of AI techniques, especially machine learning and computer vision, can be used in a variety of technological and industrial applications. Workers who acquire AI skills can have their skills cross-functional and cross-sectoral, which will help them withstand industry-specific shocks. Nevertheless, the complementarity results also suggest that AI expertise alone might not be the most effective strategy; rather, the combination of AI skills and domain-specific knowledge in fields such as semiconductors, telecommunications, or healthcare can represent the most effective career positioning.

To industry, our findings suggest that AI innovations of the past might not be rapidly replaced by newer innovations, but instead continue to be relevant by being used in other fields. This cross-domain robustness can provide more incentives to invest in AI, even in the context of creative destruction, than in traditional technologies- according to Schumpeterian growth theory [11–13]. Industries with high technological turnover can discover that long-term AI investment has compounding returns as AI capabilities allow successive rounds of complementary innovation.

These findings highlight to society and policymakers the potential of AI as a potentially transformative infrastructure that needs proper policy support. The extensive technological impact that we have recorded in our generality and

complementarity indicators implies that AI innovations have positive spillovers in a wide range of technological fields. This property is the reason why policy interventions like R&D subsidies, investments in education, and infrastructure support are justified when the technologies have a smaller scope of influence. Nevertheless, policymakers must also acknowledge the heterogeneity within AI domains-policies that encourage exploratory AI research (e.g., Machine Learning, which is most general) can have different returns than policies that encourage applied AI development (e.g., Vision, which is most complementary). The implications of international competitiveness are also worth considering: our results show that a large Asian technology company was able to build AI capabilities in several categories in 30 years, which resulted in high-impact, widely influential innovations, which indicates that long-term investment in AI research can bring considerable technological benefits irrespective of the geographic location or the initial location in the global innovation hierarchies.

### 5.4. Boundary conditions and limitations

Several important limitations condition the interpretation and generalizability of our findings. Our analysis focuses on a single firm—Samsung Electronics—operating primarily in the electronics and telecommunications industries. While the longitudinal depth spanning three decades provides rich insight into how one major technology firm developed and deployed AI capabilities, we cannot directly generalize these patterns to other organizational or industrial contexts. Startups, pure software firms, and organizations in other industries may exhibit different patterns of AI innovation and integration. Samsung's characteristics as a large, diversified, established firm with substantial R&D resources may enable AI applications and integration patterns unavailable to smaller or more specialized organizations. The complementarity patterns we observe may partly reflect Samsung's broad technological scope rather than AI's inherent characteristics.

The Korean institutional context provides additional specificity. Samsung operates within Korea's innovation system, characterized by strong government R&D support, particular industry structures, and specific patterns of university-industry collaboration. AI innovation patterns in other national contexts, particularly those with different industrial organization or innovation policies, may differ substantially. However, we note that our matching approach controls for many patent-level characteristics that might vary systematically across contexts, and our focus on USPTO patents provides some standardization.

Methodologically, our reliance on patent-based measures introduces inherent limitations. Patents capture formally claimed inventions but may not fully reflect actual technology deployment, diffusion, or commercial impact. The complementarity measure, while providing novel insight into systematic co-occurrence patterns, cannot directly reveal the mechanisms underlying these relationships or confirm actual technology integration versus coincidental classification overlap. Forward citations measure technological influence on subsequent patents but not market value, social impact, or productivity effects. The generality measures capture breadth of citing patents' technological domains but cannot determine whether this breadth reflects genuinely diverse applications or superficial overlap.

Our analysis period ending in 2018 precedes recent dramatic advances in deep learning and large language models that have transformed AI's capabilities and applications. The AI technologies in our sample primarily reflect machine learning, computer vision, and knowledge representation approaches dominant during 1987–2018. Whether newer AI approaches exhibit similar or different patterns of impact, generality, and complementarity remains an open empirical question. The rise of foundation models and their deployment across dramatically diverse applications might suggest even stronger complementarity patterns, but this conjecture requires empirical verification.

The coarsened exact matching approach, while strengthening causal inference relative to unmatched comparisons, cannot account for unobserved heterogeneity. Our matching on patent characteristics, technology fields, and temporal factors controls for many potential confounds, but unmeasured factors correlated with both AI designation and outcomes could bias our estimates. We cannot definitively rule out that some unmeasured aspect of patent quality, strategic importance, or examiner behavior drives the patterns we observe.

Finally, our focus on within-firm comparisons cannot address whether Samsung's AI innovations exhibit different patterns than AI innovations by other firms. The matching approach compares AI versus non-AI patents within Samsung's portfolio, not Samsung's AI patents versus other firms' AI patents. Cross-firm comparisons might reveal important heterogeneity in how different organizations develop and deploy AI technologies.

## 5.5. Future research directions

These limitations suggest several promising directions for future research.

Multi-firm comparative studies would illuminate whether the patterns we document are specific to Samsung or generalizable across organizations. Comparing AI innovation characteristics across firms varying in size, industry, national context, and organizational form could reveal boundary conditions and moderating factors. Particular value might come from comparing established firms like Samsung with AI-native startups to assess whether organizational heritage shapes AI innovation patterns.

Industry-specific analyses could examine whether AI's GPT-like characteristics manifest similarly across different application contexts. AI's impact, generality, and complementarity in pharmaceuticals, for example, might differ substantially from patterns in electronics given different R&D processes, regulatory environments, and innovation cycles.

Extending the temporal scope to include post-2018 AI developments would capture recent transformative advances in deep learning, natural language processing, and multimodal models. These newer AI approaches may exhibit different patterns than earlier techniques, and tracking how GPT characteristics evolve as technologies mature could provide insight into GPT emergence and development processes.

Linking patent-based measures to actual technology deployment, commercial outcomes, and productivity effects would validate whether the technological influence we document through forward citations and co-occurrence patterns translates into real economic impact. Combining patent data with product introductions, financial performance, or productivity measures could illuminate the mechanisms through which AI's technological characteristics generate value.

Cross-national comparisons could assess whether institutional contexts moderate AI innovation patterns. Comparing AI patents filed by firms from different countries, or comparing patterns across patent offices (USPTO versus EPO versus national offices), could reveal how national innovation systems shape the development and deployment of potentially transformative technologies.

This research provides foundational evidence characterizing AI innovation's distinctive technological properties within a major global technology firm. As AI continues to evolve and diffuse across industries and applications, understanding its characteristics as potentially transformative infrastructure becomes increasingly important for both innovation strategy and policy. Our findings suggest that AI's influence extends beyond narrow technical domains to enable broad technological progress and systematic integration—characteristics that justify its treatment as foundational technology deserving sustained investment and strategic attention.

## Supporting information

**S1 Dataset. Patent dataset underlying the findings of this study.**
(DOCX)

**S1 Data. dataset_Samsung AI patent analysis_20251111.**
(ZIP)

## Author contributions

**Conceptualization:** Sangrok Lee, Taehyun Jung.

**Data curation:** Taehyun Jung.

**Formal analysis:** Sangrok Lee, Taehyun Jung.

**Methodology:** Taehyun Jung.

**Project administration:** Taehyun Jung.

**Supervision:** Taehyun Jung.

**Validation:** Taehyun Jung.

**Writing – original draft:** Sangrok Lee, Taehyun Jung.

**Writing – review & editing:** Sangrok Lee, Taehyun Jung.

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
