## [Decision Letter · Decision Letter 0]

25 Sep 2025

PONE-D-25-42932Artificial Intelligence as a General-Purpose Technology? Insights from Samsung’s Patents

PLOS ONE

Dear Dr. Jung,

Thank you for submitting your manuscript to PLOS ONE. After careful consideration, we feel that it has merit but does not fully meet PLOS ONE’s publication criteria as it currently stands. Therefore, we invite you to submit a revised version of the manuscript that addresses the points raised during the review process.

Both reviewers feel that your manuscript has some potential but requires further clarifications and revisions. While you should address all comments the reviewers made, the following aspects must be thoroughly addressed in a revision:

One of my main concerns is that I am not fully convinced that the framing of general-purpose technologies match your empirical analysis. The fact that AI-patents received more citations and are more widely applicable does not necessarily make them general purpose technologies. More clarification on this essential aspect is required.I agree with Reviewer 1 that your theory section requires a revision. However, my main concern is not the age of the references but rather the disintegration of different arguments. It is, for instance, not fully clear whether and how absorptive capacity matters for your study.In this context, it is also required to embed the development of hypotheses much better in the literature (instead of having them as stand-alone items). More work needs to be done to connect your literature review to the hypotheses.The concluding and discussion section is underdeveloped and needs thorough revision and a more in-depth discussion of the implications of your work.I request to strengthen the argument about why you look at the patents of a single company. Most arguments that you bring up could be easily addressed in an econometric study with company dummies or fixed effects.Given the results in Table 5, I am really wondering what is driving the results for the generality indices in Table 4.You need to define what an AI patent is early on. Are the Ai patents on page 10 also identified using that database that is mentioned later on?What is the number of observations in the regression tables, the descriptive statistics, and the correlations?

In addition to addressing these comments, I expect you to address the Reviewer comments very carefully. Please do also take care that all references are shown in the revised version and that the correlation table contains the variable names.

If applicable, we recommend that you deposit your laboratory protocols in protocols.io to enhance the reproducibility of your results. Protocols.io assigns your protocol its own identifier (DOI) so that it can be cited independently in the future. For instructions see: https://journals.plos.org/plosone/s/submission-guidelines#loc-laboratory-protocols. Additionally, PLOS ONE offers an option for publishing peer-reviewed Lab Protocol articles, which describe protocols hosted on protocols.io. Read more information on sharing protocols at . Additionally, PLOS ONE offers an option for publishing peer-reviewed Lab Protocol articles, which describe protocols hosted on protocols.io. Read more information on sharing protocols at https://plos.org/protocols?utm_medium=editorial-email&utm_source=authorletters&utm_campaign=protocols..

We look forward to receiving your revised manuscript.

Kind regards,

Bastian Rake

Academic Editor

PLOS ONE

3. In the online submission form, you indicated that [All data files are available from the authors' personal webpage].

Additional Editor Comments (if provided):

Reviewers' comments:

Reviewer's Responses to Questions

**Comments to the Author**

1. Is the manuscript technically sound, and do the data support the conclusions?

Reviewer #1: Yes

Reviewer #2: Partly

2. Has the statistical analysis been performed appropriately and rigorously? 

Reviewer #1: Yes

Reviewer #2: Yes

3. Have the authors made all data underlying the findings in their manuscript fully available?

Reviewer #1: Yes

Reviewer #2: Yes

4. Is the manuscript presented in an intelligible fashion and written in standard English?

Reviewer #1: Yes

Reviewer #2: No

5. Review Comments to the Author

Reviewer #1: The paper entitled “Artificial Intelligence as a General-Purpose Technology? Insights from Samsung’s Patents” has an interesting topic based on using AI and modelling. But in order to be proposed for publication, the authors need to make the following improvements:

1.The Introduction is well done, an dis following the rules of writing it.

2.The THEORETICAL BACKDROP & HYPOTHESIS DEVELOPMENT is suffering of adding new and updated sources. As the authors use a topic and analyse it under the impact of AI, the authors need to use upadated references. They need to improve this weaknesses as follows: at every sub-section from the second section, the authors need to add new and updated sources especially from 2023-2025 (they used sources from 1990, 2001, 2010, 2020). So is very important, not only for the journal, but also for the readers.

3.The Methodology is well done.

4.Contribution to Literature must be improved. The authors used only one source (from 1995), so to measure the gap between the study and the literature in the filed, they have to use a few new and updated sources.

5.Practical implications need also a more carefull description. It is indicated to describe PI on categories (write them using italic writing for sub-titles): for employees, for managers, for the industry, for society. Being such a challenging and dynamic field, it is requiered such a description based on results and innovation.

6.References must be updated, as the requirement, being a dynamic field under the impact of AI.

Having these proposals in view, the paper, receives minor revision.

Reviewer #2: Dear Authors,

Thank you very much for giving me the possibility to read your paper. I organized the review as follows: the first part with the Summary and Contribution, and the Overall Assessment. Major and minors comments follow.

Summary and Contribution

This manuscript compares AI-related patents to non-AI patents within the same firm, evaluating whether AI patents exhibit features consistent with General-Purpose Technologies (GPTs). Using coarsened exact matching (CEM) and count models (NB/ZINB) for forward citations, plus censored models (Tobit/Heckman) for generality (Jaffe–Trajtenberg–Henderson and Shannon measures), the paper reports higher impact and higher generality for AI patents. The within-firm design is a clear strength that reduces organizational heterogeneity and supports a focused interpretation.

Overall Assessment

The study is promising and mostly well designed. However, several theoretical and methodological clarifications, robustness checks, and presentation issues currently limit the strength and generalizability of the conclusions. I recommend major revisions aimed at (i) better framing the results in a firm-level context, avoiding over-generalization; (ii) tightening the empirical strategy and reporting (especially for CEM, self-citations, and standard errors); and (iii) correcting formatting and language issues.

Major Comments

1. Scope and Generalizability

Findings are compelling within the focal firm. The broader claim that “AI exhibits GPT-like features” should be qualified as firm-level evidence unless corroborated with additional firms or an external benchmark. Please moderate the language in the Abstract/Conclusion accordingly and discuss limits to external validity. For instance, the sentence “Our findings provide strong empirical support for the view of AI as a GPT” (p. 33) reads as too strong relative to the study’s design.

2. CEM: Weights, Balance, and Sensitivity

Clarify whether CEM weights were used in the outcome models and report post-matching balance (means/SDs/standardized differences) beyond inventors/claims—e.g., CPC breadth, backward citations (patent/NPL), WIPO field distribution. Add concise sensitivity analyses:

- 1:1 vs many-to-many matching;

- Tighter/looser calipers on filing year;

- Tests for potential over-matching on CPC overlap (which may partially mediate generality).

Consider showing results both with and without strict CPC-overlap requirements.

3. Forward Citations: Self-Citations and Assignee of Citing Patents

State explicitly how self-citations are handled. At minimum, include robustness checks excluding assignee self-citations and/or control for their share. Where feasible, account for the assignee of citing patents to ensure the observed impact is not driven by internal reuse only.

4. Field/Time Normalization and Additional Controls

Although you include fixed effects, consider field- and year-normalized citation indicators (e.g., percentile ranks within filing-year × field) and alternative windows (3/5/7-year) to test stability. Add controls for family size, grant lag, and—if available—examiner art-unit or technology-specific grant propensities.

5. Inference: Standard Errors and Model Variants

Report cluster-robust standard errors (e.g., by filing year and/or technology field; two-way if justified). As an additional check, compare ZINB with a zero-truncated NB on the positive-citations subsample and report average marginal effects with CIs for interpretability—this isolates the intensity margin, ensuring the AI effect is not driven solely by zero inflation.

6. Heterogeneity Analyses

Your sub-technology analyses are intriguing. Please present marginal effects/AME with confidence intervals across AI sub-domains and clarify which effects remain significant once controlling for impact (to separate breadth from magnitude).

Minor Comments (Language, Formatting, Structure, Clarity)

- Prefer plainer wording to some unusual expressions (e.g., “precise economic characterization” at the beginning of p. 2). On p. 5, clarify what “chemical space” means to a general audience.

- In 2. Theoretical Background & Hypothesis Development, avoid overly short subsections (e.g., three subsections on a single page). A similar point applies to 5. Discussion & Conclusion; merging some subsections would improve flow.

- In 3.6 Estimation Models, the first forward-citation equation is not accompanied by a variable legend, unlike the second (generality) equation—please add a compact legend for consistency.

- Still in 3.6 Estimation Models, you write: “For completeness, we also ran simple OLS models… These results are omitted for brevity.” Either provide them in an Appendix or omit the sentence.

- Enrich 5.2 Contributions to the Literature (currently it cites only one paper), to better position the contribution.

- Remove all placeholder cross-reference errors such as “Error! Reference source not found.” and ensure figure/table numbering is consistent.

- Tighten a few long sentences and check punctuation (missing commas/parentheses) to meet standard English requirements.

- For ZINB/Tobit/Heckman tables, consider adding a pseudo-R² (where meaningful) and a compact panel of marginal effects (AME) with CIs to translate coefficients into interpretable units.

6. PLOS authors have the option to publish the peer review history of their article (what does this mean?). If published, this will include your full peer review and any attached files.). If published, this will include your full peer review and any attached files.

.

Reviewer #1: No

Reviewer #2: No

While revising your submission, please upload your figure files to the Preflight Analysis and Conversion Engine (PACE) digital diagnostic tool, https://pacev2.apexcovantage.com/. PACE helps ensure that figures meet PLOS requirements. To use PACE, you must first register as a user. Registration is free. Then, login and navigate to the UPLOAD tab, where you will find detailed instructions on how to use the tool. If you encounter any issues or have any questions when using PACE, please email PLOS at . PACE helps ensure that figures meet PLOS requirements. To use PACE, you must first register as a user. Registration is free. Then, login and navigate to the UPLOAD tab, where you will find detailed instructions on how to use the tool. If you encounter any issues or have any questions when using PACE, please email PLOS at figures@plos.org. Please note that Supporting Information files do not need this step.. Please note that Supporting Information files do not need this step.

---

## [Decision Letter · Decision Letter 1]

26 Dec 2025

PONE-D-25-42932R1Technological Impact, Generality, and Complementarity of Artificial Intelligence Patents: Evidence from Samsung ElectronicsPLOS One

Dear Dr. Jung,

Thank you for submitting your manuscript to PLOS ONE. After careful consideration, we feel that it has merit but does not fully meet PLOS ONE’s publication criteria as it currently stands. Therefore, we invite you to submit a revised version of the manuscript that addresses the points raised during the review process.

If applicable, we recommend that you deposit your laboratory protocols in protocols.io to enhance the reproducibility of your results. Protocols.io assigns your protocol its own identifier (DOI) so that it can be cited independently in the future. For instructions see: https://journals.plos.org/plosone/s/submission-guidelines#loc-laboratory-protocols. Additionally, PLOS ONE offers an option for publishing peer-reviewed Lab Protocol articles, which describe protocols hosted on protocols.io. Read more information on sharing protocols at . Additionally, PLOS ONE offers an option for publishing peer-reviewed Lab Protocol articles, which describe protocols hosted on protocols.io. Read more information on sharing protocols at https://plos.org/protocols?utm_medium=editorial-email&utm_source=authorletters&utm_campaign=protocols..

We look forward to receiving your revised manuscript.

Kind regards,

Bastian Rake

Academic Editor

PLOS One

Journal Requirements:

Additional Editor Comments:

While the manuscript has considerably improved, the Reviewers – and particularly Reviewer 1 – raise important concerns regarding the need for further engagement with the literature. I support Reviewer 1’s view and ask you to address these concerns very thoroughly, but also to address the point Reviewer 2 has made.

While extending the engagement with the literature, will require some work, it will strengthen your arguments and the quality of your study.

Reviewers' comments:

Reviewer's Responses to Questions

**Comments to the Author**

1. If the authors have adequately addressed your comments raised in a previous round of review and you feel that this manuscript is now acceptable for publication, you may indicate that here to bypass the “Comments to the Author” section, enter your conflict of interest statement in the “Confidential to Editor” section, and submit your "Accept" recommendation.

Reviewer #1: (No Response)

Reviewer #2: All comments have been addressed

2. Is the manuscript technically sound, and do the data support the conclusions?

Reviewer #1: Partly

Reviewer #2: Yes

3. Has the statistical analysis been performed appropriately and rigorously? 

Reviewer #1: Yes

Reviewer #2: Yes

4. Have the authors made all data underlying the findings in their manuscript fully available?

Reviewer #1: Yes

Reviewer #2: Yes

5. Is the manuscript presented in an intelligible fashion and written in standard English?

Reviewer #1: Yes

Reviewer #2: Yes

6. Review Comments to the Author

Reviewer #1: Response 20.12.2025

As it was added before the paper entitled “Artificial Intelligence as a General-Purpose Technology? Insights from Samsung’s Patents” has an interesting topic based on using AI and modelling. But in order to be proposed for publication, the authors need to make the following improvements:

2.The THEORETICAL BACKDROP & HYPOTHESIS DEVELOPMENT is suffering of adding new and updated sources. As the authors use a topic and analyse it under the impact of AI, the authors need to use upadated references. Only one source per described section it does not indicating the improvement according to the rquests.

The authors must ad t least 2-3 sources especially from 2025, duet o the dynamism of the digital impact.

4.Contribution to Literature must be improved. The authors used only one source (from 1995), so to measure the gap between the study and the literature in the filed, they have to use a few new and updated sources.

The authors must ad a few updated sources from 2025.

5.Practical implications need also a more carefull description. It is indicated to describe PI on categories (write them using italic writing for sub-titles): for employees, for managers, for the industry, for society.

The authors added just implications for firms and policymakers, the other requested implications are still missing. More careful attention, please!

6.References must be updated, as the requirement, being a dynamic field under the impact of AI.

The authors must add a few more sources especially from 2025!!!

Having these proposals in view, the paper receives major revision, so please pay attention to the indicated improvement measures!

Reviewer #2: Dear Authors,

Thank you for having thoroughly and correctly addressed all the comments. Now, the revised manuscript is substantially improved and addresses many of the key concerns raised by the editor and reviewers. In particular, the framing is now more appropriately qualified as firm-level evidence, the definition/identification of AI patents is clearer, and the empirical section is strengthened through additional robustness checks (e.g., alternative specifications and windows, treatment of self-citations, normalization choices). The introduction of “complementarity” as an additional GPT-related dimension is a meaningful step that helps align the empirical analysis more closely with the conceptual discussion of GPT characteristics.

That said, I still see a few substantive issues that should be still addressed:

1. GPT framing and strength of claims. While the new version moderates the language compared with the prior version, the manuscript should remain fully consistent throughout (abstract, discussion, conclusion) in presenting the findings as evidence consistent with GPT-like characteristics within the focal firm, rather than as a broad characterization of AI as a GPT in general. For example, in the 1.Introduction the text states that AI’s properties are “defining features of GPTs.”

Suggestion: Rephrase such statements to make clear that these are GPT-like features tested in this study within a focal firm context. For example, replace wording like “defining features” with “often discussed as GPT-like features and assessed here using patent indicators within a single firm context.”

2. Interpretation of complementarity. The new measure is interesting and well motivated, but some statements interpret the results in terms of “purposeful/deliberate” technological coupling. Given the observational nature of the study, I encourage the authors to soften the intentional language and present the results as systematic patterns consistent with complementarity, avoiding claims about intent that the data cannot directly observe. Your measure shows systematic co-occurrence beyond random expectation (a strong and useful result), but it does not directly observe managerial intent.

Suggestion: Replace “purposeful/intentional/deliberate” with “systematic / non-random / consistent with complementarity”.

3. Connecting results to “what drives generality”. The paper would benefit from a clearer explanation of what is driving differences in the generality indices (especially given the discussion around controls and related results), Indeed, the main text (especially the Discussion/Conclusion) still does not explicitly answer the editor’s question (comment 5) in a single, easy-to-spot takeaway. Readers may still ask: is higher generality mainly because AI patents are more cited, or because they are genuinely broader even conditional on citations? You already have language and analyses that can help answer this (e.g., the “total vs pure generality”).

Suggestion: Add a short paragraph in the 5. Discussion and conclusion section that explicitly states (in plain terms) what your results imply. This would also help readers understand whether generality differences reflect breadth per se, citation intensity, or other correlated patent characteristics.

Overall, the new version represents a clear step forward. With these final conceptual/interpretative adjustments—especially around consistent GPT framing and cautious interpretation of complementarity/generality—the manuscript would be much stronger.

7. PLOS authors have the option to publish the peer review history of their article (what does this mean?). If published, this will include your full peer review and any attached files.). If published, this will include your full peer review and any attached files.

.

Reviewer #1: No

Reviewer #2: No

---

## [Author Response · Author response to Decision Letter 2]

13 Feb 2026

Please see the attached file of the response to reviewers

---

## [Decision Letter · Decision Letter 2]

16 Mar 2026

Technological Impact, Generality, and Complementarity of Artificial Intelligence Patents: Evidence from Samsung Electronics

PONE-D-25-42932R2

Dear Dr. Taehyun Jung,

We’re pleased to inform you that your manuscript has been judged scientifically suitable for publication and will be formally accepted for publication once it meets all outstanding technical requirements.

An invoice will be generated when your article is formally accepted. Please note, if your institution has a publishing partnership with PLOS and your article meets the relevant criteria, all or part of your publication costs will be covered. Please make sure your user information is up-to-date by logging into Editorial Manager at Editorial Manager® and clicking the ‘Update My Information' link at the top of the page. For questions related to billing, please contact  and clicking the ‘Update My Information' link at the top of the page. For questions related to billing, please contact billing support..

Kind regards,

Zeyu Xing

Academic Editor

PLOS One

Additional Editor Comments (optional):

Reviewers' comments:

Reviewer's Responses to Questions

**Comments to the Author**

1. If the authors have adequately addressed your comments raised in a previous round of review and you feel that this manuscript is now acceptable for publication, you may indicate that here to bypass the “Comments to the Author” section, enter your conflict of interest statement in the “Confidential to Editor” section, and submit your "Accept" recommendation.

Reviewer #2: All comments have been addressed

2. Is the manuscript technically sound, and do the data support the conclusions?

Reviewer #2: Yes

3. Has the statistical analysis been performed appropriately and rigorously? 

Reviewer #2: Yes

4. Have the authors made all data underlying the findings in their manuscript fully available?

Reviewer #2: Yes

5. Is the manuscript presented in an intelligible fashion and written in standard English?

Reviewer #2: Yes

6. Review Comments to the Author

Reviewer #2: Dear Authors,

thank you for your careful and constructive revision of the manuscript. The revised version shows substantial improvement in theoretical positioning, conceptual clarity, and overall coherence. In my assessment, you have addressed all previously raised concerns thoroughly and in a satisfactory way.

First, the engagement with recent literature has been significantly strengthened. The theoretical backdrop now incorporates up-to-date contributions (2024–2025) on artificial intelligence as a general-purpose technology and on patent-based measures of technological impact, generality, and complementarity. The revised discussion puts the hypotheses within the evolving AI–GPT debate rather than relying predominantly on foundational work from the 1990s. This considerably enhances the manuscript’s timeliness and scholarly relevance.

Second, the contribution to the literature is now more clearly articulated. You convincingly distinguish your study from recent firm-level AI research that treats AI primarily as an input affecting organizational outcomes. By focusing instead on the intrinsic technological characteristics of AI patents—specifically impact, generality, and complementarity—the manuscript clarifies its distinct empirical contribution to the current research frontier.

Third, the conceptual framing has been appropriately refined. The language regarding GPT claims has been moderated throughout the manuscript. The findings are now presented as evidence of GPT-like characteristics within a focal firm context, rather than as definitive claims about AI universally constituting a general-purpose technology. This increased precision strengthens the theoretical integrity of the study.

Fourth, the interpretation of complementarity has been improved. The revised manuscript removes language implying intentional or strategic technological coupling that could not be directly inferred from the observational design. Complementarity is now described as systematic, non-random co-occurrence patterns, which is methodologically sound and interpretively appropriate.

Fifth, the added clarification regarding the mechanisms underlying generality—distinguishing citation intensity effects from genuine cross-domain technological breadth—addresses a potentially important concern and reinforces the robustness of the results.

Finally, the practical implications section has been thoughtfully restructured. By organizing implications according to stakeholder categories (managers, employees, industry, and society), the authors provide a clearer and more actionable translation of their empirical findings, enhancing the manuscript’s broader relevance.

Overall, the manuscript presents a well-constructed dataset, a rigorous empirical strategy—including matching procedures and multiple model specifications—and consistent, robust findings that are carefully interpreted. The revised version is theoretically grounded, methodologically sound, and substantively relevant to ongoing discussions about the technological nature of AI.

I therefore support publication of the manuscript in its current form.

7. PLOS authors have the option to publish the peer review history of their article (what does this mean?). If published, this will include your full peer review and any attached files.). If published, this will include your full peer review and any attached files.

.

Reviewer #2: No

---

## [Editor Report · Acceptance letter]

PONE-D-25-42932R2

PLOS One

Dear Dr. Jung,

I'm pleased to inform you that your manuscript has been deemed suitable for publication in PLOS One. Congratulations! Your manuscript is now being handed over to our production team.

Kind regards,

on behalf of

Dr. Zeyu Xing

Academic Editor

PLOS One